# PRMT5 inhibition disrupts splicing and stemness in glioblastoma

Patty Sachamitr[1,2], Jolene C. Ho [2], Felipe E. Ciamponi [3,4], Wail Ba-Alawi[5,6], Fiona J. Coutinho[1], Paul Guilhamon[1,5], Michelle M. Kushida[1], Florence M. G. Cavalli[1], Lilian Lee[1], Naghmeh Rastegar[1], Victoria Vu[2,5], María Sánchez-Osuna [7], Jasmin Coulombe-Huntington[7], Evgeny Kanshin[7], Heather Whetstone[1], Mathieu Durand[8], Philippe Thibault[8], Kirsten Hart[2,6], Maria Mangos[2], Joseph Veyhl[2], Wenjun Chen[2], Nhat Tran[2], Bang-Chi Duong[2], Ahmed M. Aman [9], Xinghui Che[1], Xiaoyang Lan [1], Owen Whitley[10,11], Olga Zaslaver[10,11], Dalia Barsyte-Lovejoy [2,12], Laura M. Richards [5,6], Ian Restall [13,14], Amy Caudy [10,11,15], Hannes L. Röst [10,11], Zahid Quyoom Bonday[16], Mark Bernstein[17,18], Sunit Das [17,19,20], Michael D. Cusimano[19], Julian Spears[17,21], Gary D. Bader [10,11], Trevor J. Pugh [5,6,9], Mike Tyers[7], Mathieu Lupien [5,6,9], Benjamin Haibe-Kains[5,6,9,22,23], H. Artee Luchman[13,14,24], Samuel Weiss[13,14,24], Katlin B. Massirer[3,4], Panagiotis Prinos [2✉], Cheryl H. Arrowsmith [2,5,6✉] & Peter B. Dirks [1,11,17,20,25,26✉]

Glioblastoma (GBM) is a deadly cancer in which cancer stem cells (CSCs) sustain tumor growth and contribute to therapeutic resistance. Protein arginine methyltransferase 5 (PRMT5) has recently emerged as a promising target in GBM. Using two orthogonal-acting inhibitors of PRMT5 (GSK591 or LLY-283), we show that pharmacological inhibition of PRMT5 suppresses the growth of a cohort of 46 patient-derived GBM stem cell cultures, with the proneural subtype showing greater sensitivity. We show that PRMT5 inhibition causes widespread disruption of splicing across the transcriptome, particularly affecting cell cycle gene products. We identify a GBM splicing signature that correlates with the degree of response to PRMT5 inhibition. Importantly, we demonstrate that LLY-283 is brain-penetrant and significantly prolongs the survival of mice with orthotopic patient-derived xenografts. Collectively, our findings provide a rationale for the clinical development of brain penetrant PRMT5 inhibitors as treatment for GBM.

---

A list of author affiliations appears at the end of the paper.

The prognosis for primary glioblastoma (GBM) patients remains dismal, with <10% surviving beyond 5 years[1]. The current standard of care chemotherapy, oral alkylating agent temozolomide, is only effective in a subset of patients and extends the median survival of patients by merely 3 months[1]. The poor outcome of GBM could be, at least partially, attributed to the presence of a tumor-initiating cell population, known as cancer stem cells (CSCs)[2], which have been shown to drive tumor growth and resistance to therapy[3–5]. It follows that effective tumor eradication will require novel therapeutic strategies that target CSCs in addition to bulk tumor cells.

Epigenetic regulation has been identified as a key player in the development of many cancers, including gliomas[6,7]. Indeed, 46% of GBM patients harbor at least one mutation in genes linked to chromatin organization[8], suggesting that epigenetic processes may drive GBM across heterogeneous molecular subtypes. These findings suggest that agents that modify these processes may find application in GBM, particularly given that these tumors demonstrate strong stemness features that crosscut mutational diversity between patients.

Protein arginine methyltransferase 5 (PRMT5) has recently emerged as a promising target in many cancers[9]. It catalyzes the majority of symmetric arginine dimethylation in many histone and non-histone proteins, including p53 and epidermal growth factor receptor[9,10]. PRMT5 has recently been linked with the maintenance of self-renewal in leukemic stem cells[11] and shown to be critical for breast CSC function[12]. Expression of PRMT5 correlates with grade of malignancy in gliomas and inversely correlates with survival[10,13]. Genetic suppression[10,13–15] and pharmacological inhibition of PRMT5 in mouse, zebrafish, and human GBM models both result in growth suppression, providing a rationale for further investigation of this druggable target on the tumorigenic subpopulation[14,16,17].

To identify epigenetic vulnerabilities in GBM, we screened a collection of patient-derived glioblastoma CSC (GSC) lines with a focused library of validated chemical probes targeting epigenetic regulators[16,17]. We identified PRMT5 inhibitors GSK591 and LLY-283 each with different chemotypes and modes of action as potent suppressors of patient-derived GSCs. Given the inter-patient heterogeneity of GBM patient tumors, we investigated PRMT5 as a target in a dose–response series across an expanded set of 46 well-characterized patient-derived GSC lines. PRMT5 inhibition is effective in attenuating the growth and clonogenic capacity of a large proportion of patient-derived GSCs, and the central nervous system (CNS)-penetrant chemical probe LLY-283 provides a significant survival benefit in an orthotopic patient-derived xenograft model. PRMT5 inhibition leads to dramatic changes in mRNA splicing and predominantly increased exon skipping and intron retention that disrupt the levels of transcripts involved in cell cycle progression. These data provide a strong rationale for the further development of PRMT5 inhibitors for clinical application in the treatment of GBM patients.

## Results
### Epigenetic chemical probe screen identifies PRMT5 as a potential therapeutic target for GBM.
We evaluated the effects of a library of 39 well-characterized epigenetic chemical probes in a proliferation screen of 26 patient-derived GSC lines (grown adherently or as spheres) derived from adult primary GBM. Three human fetal neural stem (HFNS) cell lines were also screened as a practical and relevant control population of normal cells that are grown under identical conditions. The screen readout was a measure of confluence, as determined by real-time live cell imaging. The probes represent a collection of small molecules that selectively and potently inhibit methyltransferases,

bromodomains, acetyltransferases, and deacetylases (Supplementary Table 1). Three chemical probes, JQ1, GSK591, and LLY-283, were found to significantly inhibit GSC and HFNS cell proliferation by >50% in more than half the GSC lines screened compared to the vehicle control (dimethyl sulfoxide (DMSO; Fig. 1a). JQ1, an inhibitor of the bromodomain and extra-terminal family of proteins, has been widely demonstrated to have growth-inhibitory properties on a wide variety of cancer cells, including glioma[18]. As PRMT5 inhibitors were hits across many patient-derived GSCs, we sought to further investigate PRMT5 as a therapeutic target in GBM and in a larger cohort of primary patient-derived cell models to determine the potential breadth of its efficacy. GSK591 and LLY-283 are chemically unrelated inhibitors of PRMT5 methyltransferase activity (Fig. 1b); GSK591 is a substrate-competitive inhibitor of the PRMT5-MEP50 complex[19], while LLY-283 is a cofactor-competitive inhibitor that binds to the S-adenosyl methionine-binding site of PRMT5[20].

### PRMT5 inhibition impairs the proliferation and sphere-forming capacity (SFC) of GSCs and primary GBM cells.
To validate the effect of PRMT5 inhibition on patient-derived GSCs, we performed a more detailed study of the response to PRMT5 inhibition in three representative GSC lines, G561, G411, and G583. Each line was treated with nine different concentrations of GSK591 or LLY-283 and cultured until the untreated controls reached confluence. All three lines were found to be sensitive to PRMT5 inhibition in a dose-dependent fashion, with EC50 values in the low nanomolar range (10–19 nM) for LLY-283 and low micromolar range (0.1–1 μM) for GSK591 (Figs. 1c, d). We also followed the growth of these cells using real-time live-cell imaging for 9–12 days after treatment with LLY-283, GSK591, or its inactive control SGC2096. Both PRMT5 inhibitors, but not the inactive control, had a profound effect on the growth of the three GSC lines assayed, with LLY-283 having greater potency compared to GSK591 (Supplementary Fig. 1a). To investigate whether PRMT5 inhibition affected normal brain cells, we performed a similar assay on normal human astrocytes (NHAs), which showed minimal response to PRMT5 inhibition (Supplementary Fig. 1b). However, HFNS cells were found to be sensitive to PRMT5 inhibition, displaying a range of EC50s similar to GSCs, suggesting that these compounds have activity on CNS cells that have stem cell properties (Supplementary Fig. 1c). As pediatric GBMs have distinct molecular differences relative to adult GBM, we also tested PRMT5 inhibition on three pediatric GSC lines, G477, G626, and G752, using similar dose–response assays. We found that pediatric GSCs were also sensitive to PRMT5 inhibition with EC50 values in the low nanomolar range for LLY-283 and low micromolar range for GSK591 (Supplementary Fig. 1d).

It is known that PRMT5 is responsible for the majority of the symmetric dimethylation of arginine (SDMA) in cells[21]. To confirm that both PRMT5 inhibitors were effective in inhibiting PRMT5 activity, we treated three representative GSC lines with 1 μM GSK591 or LLY-283 for 5 days and measured the levels of SDMA on the PRMT5 target SmB/B' protein using western blot[22]. A strong reduction of the SDMA mark was observed in all three GSC lines treated with 1 μM GSK591 or LLY-283 but not with the inactive control compound SGC2096 (Fig. 1e). Fetal neural stem cells (HFNS) treated with 1 μM GSK591 or LLY-283 also showed significant decreases in SDMA levels as did NHAs (Supplementary Fig. 1e) confirming that both chemical probes inhibit PRMT5 catalytic activity in GSCs as well as normal CNS cells.

To investigate whether PRMT5 inhibition affects the clonogenic properties of GSCs, a hallmark of CSCs, we conducted an in vitro limiting dilution assay (LDA) on the above mentioned

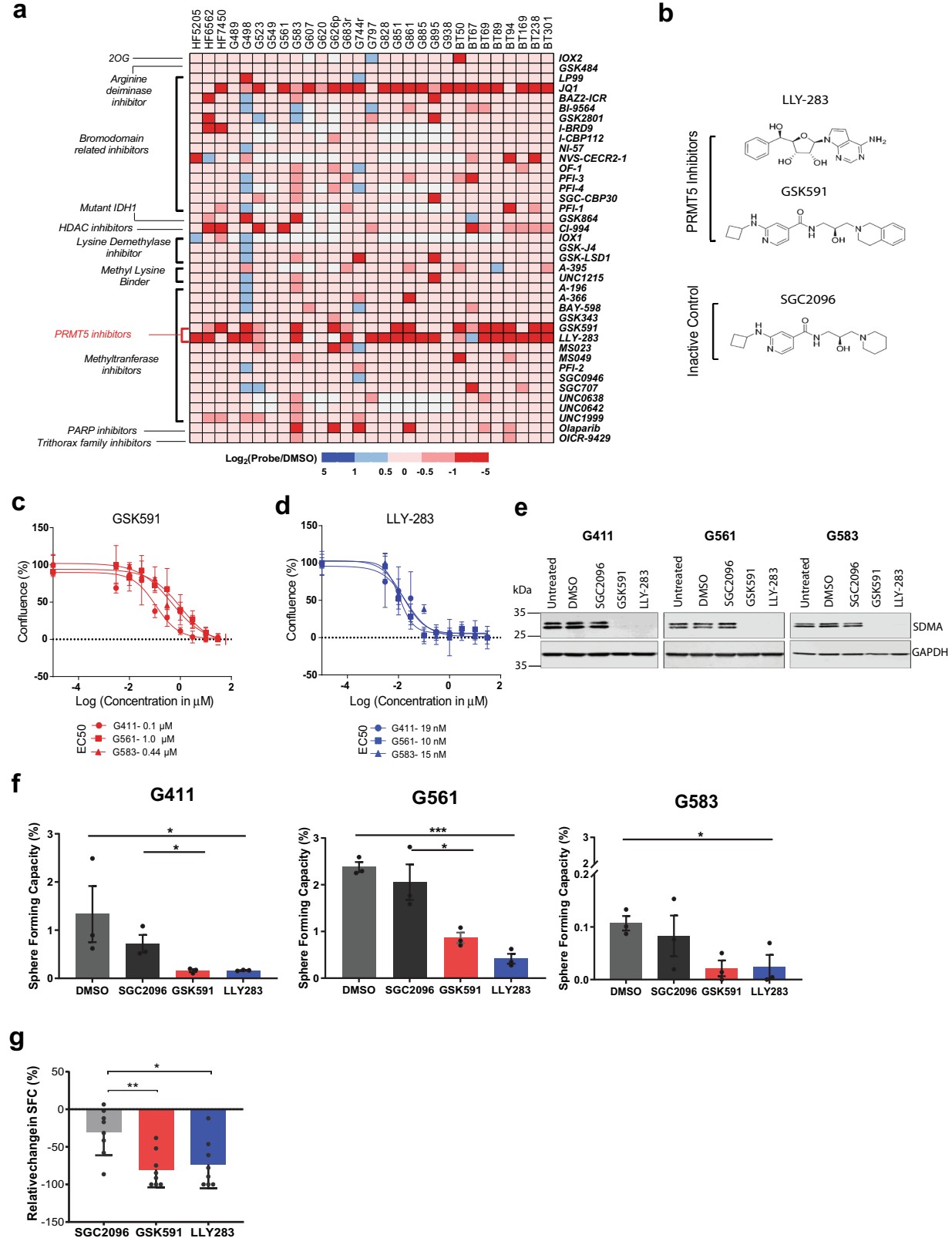

three representative GSC lines. Although the baseline SFC varied across patient-derived GSCs, we observed a significant reduction in the SFC across all three GSC lines when treated with 1 μM of GSK591 or LLY-283, with LLY-283 being more potent (Fig. 1f). To confirm that the effect of PRMT5 inhibition on SFC was not specific to these established GSC cultures, we further investigated

the effect of PRMT5 inhibition on the growth of freshly dissociated (passage 0) primary patient-derived GBM cells from nine patient tumors. This assay gives a measure of the growth capacity of cells as close to the fresh tumor as possible. Six of the nine primary GBM cultures studied showed significant reduction in SFC upon treatment with both PRMT5 inhibitors, suggesting

**Fig. 1 Small molecule inhibition of PRMT5 impairs both proliferation and self-renewal in GSCs. a** Cell confluence heatmap of small molecule epigenetic screen showing significant inhibition of GSC proliferation by PRMT5 inhibitors. The screen was performed on 26 GSC lines and 3 control cell lines against 39 epigenetic chemical probes at 1 µM final concentration. Cells were grown adherently for 12–14 days and scored for confluence using high-throughput live-cell imaging. Data are represented as $\log_2$ (confluence with probe/confluence with DMSO). Red squares indicate decrease in cell confluence and blue squares indicate increase in confluence relative to vehicle. White squares indicate probe and cell line combinations that were not screened. **b** Chemical structures of the PRMT5 inhibitors, GSK591 and LLY-283, alongside the inactive control, SGC2096. **c, d** Percentage confluence of three GSC lines upon treatment with GSK591 (**c**—red) and LLY-283 (**d**—blue), with doses ranging from 3 nM to 30 µM. Dose response is calculated after 9–12 days of treatment with inhibitors/controls (until control wells are confluent). Data shown are representative of three independent experiments, mean ± SD. **e** Western blots of the SDMA mark on the SmB/B′ protein in three representative GSC lines G411, G561, and G583 following 5-day treatment with 1 µM of GSK591, LLY-283, SGC2096, or DMSO control. The western blot experiments were reproduced at least three independent times using cell lysates from different biological replicates. **f** Limiting dilution analysis (LDA) performed on three GSC lines treated with 1 µM of the PRMT5 inhibitors, GSK591 and LLY-283, and controls, SGC2096 and DMSO, for 14 days. Data show the percentage of sphere-forming capacity. $n = 3$ biologically independent samples, mean ± SEM. Two-tailed Unpaired $t$ test, $p$ values: G411-GSK/SGC = 0.0396, G561-LLY/DMSO = 0.0002, G561-GSK/SGC = 0.0400, G583-LLY/DMSO = 0.0350, G583-GSK/SGC = 0.2127. **g** Summary of limiting dilution analysis (LDA) performed on freshly dissociated GBM cells from 9 patients treated with 1 µM of the PRMT5 inhibitors, GSK591 and LLY-283, and controls, SGC2096 and DMSO, for 21 days. The y-axis shows the percentage relative change in sphere-forming capacity (normalized to DMSO). $N = 9$ biologically independent patient-derived GBM samples, mean ± SEM, individual data per sample are shown in Supplementary Fig. 1f. Two-tailed unpaired $t$ test with Welch's correction. $p$ values: LLY283/SGC2096 = 0.0157, GSK591/SGC2096 = 0.0031. $*p < 0.05$; $**p < 0.01$; $***p < 0.001$. Source data are provided as a Source data file.

that PRMT5 inhibition limits the upfront clonogenic potential of the majority of fresh primary GBM cells (Fig. 1g and Supplementary Fig. 1f).

**Patient-derived GSCs across heterogeneous subtypes show differential response to PRMT5 inhibition.** Due to the well-documented patient inter-tumoral heterogeneity of GBM[5,23,24], it is difficult to faithfully capture the complex biology of this disease with only a small collection of patient-derived cells or lines. To validate PRMT5 as a target in GBM, we took advantage of a large unique and well-characterized cohort of patient-derived GSC lines grown either adherently on laminin (annotated "Gxxx"—29 lines) or grown as neurospheres (annotated "BTxxx"—17 lines). Both models have been shown to faithfully recapitulate patient tumor histology upon intracranial implantation into mice and have been extensively used to investigate numerous aspects of GBM biology[5,25,26].

We performed dose–response assays with nine concentrations (ranging from 3 nM to 30 µM) of GSK591 and LLY-283 for 46 patient-derived GSC lines and utilized a live-cell imaging platform to monitor cell confluence over time for adherent lines or Alamar blue for GSCs grown as spheres. The area above the curve (AAC) was calculated from dose–response curves to measure the efficacy of the compounds, with a higher AAC value (on a scale between zero to one) signifying greater sensitivity to PRMT5 inhibition. We observed a gradation of responses to both GSK591 and LLY-283 across the 46 patient-derived GSC lines assayed (Fig. 2a), with AACs ranging from 0 to 0.55 for GSK591 and from 0.002 to 0.83 for LLY-283 (Supplementary Fig. 2a). We observed strong correlation (Pearson's $r = 0.803$; $p < 0.0001$) between the response to GSK591 and LLY-283 across all cell lines (Supplementary Fig. 2b), with LLY-283, on average, more potent at attenuating GSC growth (Supplementary Fig. 2a). Although these data demonstrate sample-dependent variability in response to PRMT5 inhibition, more than half (56%) the GSC lines were highly sensitive to LLY-283, with EC50s below 250 nM (Supplementary Table 2).

To further investigate the differential response, we derived mutation and copy number variation (CNV) status from whole-genome sequencing data for a subset of 33 lines. These lines were found to possess the spectrum of common genomic alterations found in GBM, representative of the four gene expression-based molecular subgroups of GBM, as defined by Verhaak et al. in 2010[27] (Fig. 2b). The response to PRMT5 inhibition was

independent of genetic mutation, CNV status, or gene expression subgroup, with a spectrum of responses seen across lines (Fig. 2b). Interestingly, patient GSC samples identified as being mostly proneural, based on RNA sequencing (RNA-seq) data and TCGA criteria, were enriched within the more sensitive GSC lines (Fig. 2b), suggesting that GBM patients with more proneural subtype tumors may be more sensitive to PRMT5 inhibitors.

In previous studies, sensitivity to PRMT5 genetic ablation or chemical inhibition has been linked to *TP53* mutations and *MTAP* loss[28–30]. However, our data show no correlation of PRMT5 inhibition to either *TP53* mutation or *MTAP* copy number status (Fig. 2b). Because the *MTAP* locus overlaps with the *CDKN2A* locus (encoding p16INK4A), we investigated the sensitivity to PRMT5i with respect to protein levels of MTAP or p16INK4A as evidenced by western blots (Fig. 2c). Although there was general correlation between MTAP and CDKN2A expression, four GSC lines show discordant CNV status between *MTAP* and *CDKN2A* at the genomic level, and an additional nine GSC lines showed discordant MTAP and p16INK4A protein expression (Fig. 2b, c). This is not unexpected as it has been reported that *MTAP* can be independently deleted relative to *CDKN2A*[31]. The additional discordance at the protein level could be due to epigenetic silencing via selective promoter methylation in either gene as has been frequently reported in various cancers[32,33]. Moreover, metabolomics analysis revealed no correlation of response to PRMT5 inhibitors with intracellular or extracellular methylthioadenosine (MTA) levels (Supplementary Fig. 2c). Importantly, we also found that the response to PRMT5 inhibition did not correlate with the ratio of expression of *CLNS1A* to *RIOK1* (Supplementary Fig. 2d), which was previously reported as a biomarker of sensitivity to PRMT5 inhibition in a panel of immortalized glioma cell lines[34]. Thus previously proposed explanations for variation of response to PRMT5 inhibition do not appear to be applicable to our extensive panel of low-passage patient-derived GSC lines.

To further investigate factors that may influence the sensitivity of our GSC lines to PRMT5 inhibition, we compared the efficacy of PRMT5 chemical probes to reduce the SDMA mark in three non-responder cell lines (Fig. S2G) compared to three good responding cell lines (Fig. 1e). We confirmed that GSK591 and LLY-283 were equally efficacious in inhibiting PRMT5 enzymatic activity in all six GSC lines as measured by the levels of SDMA by western blot, thus excluding differences in drug uptake, efflux, stability, or metabolism in the non-responders vs respondents. Taken together, our data indicate that PRMT5 inhibition can

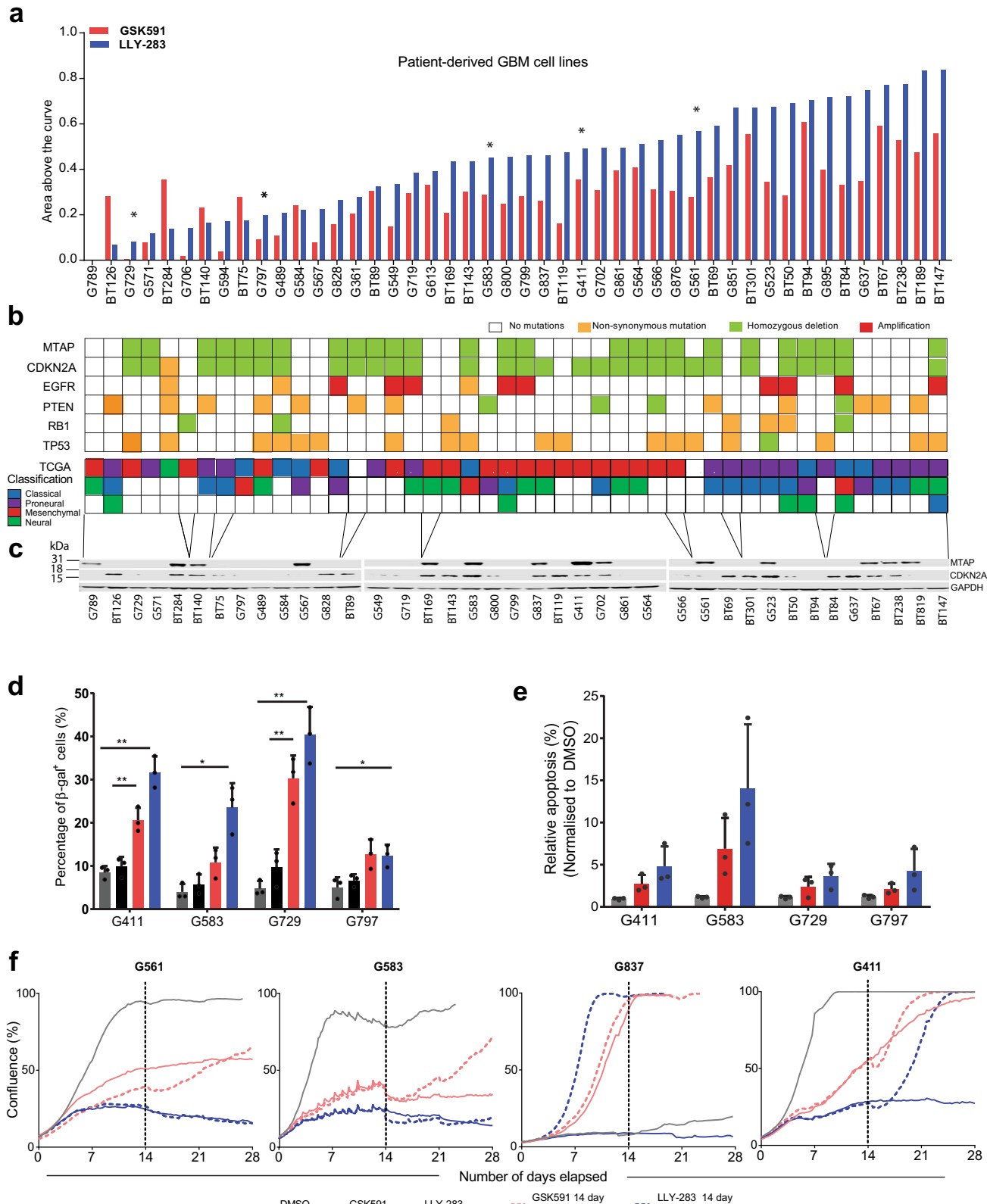

be effective in a wide variety of molecular and phenotypic subtypes, which is a desirable property for the treatment of this heterogeneous disease.

**PRMT5 inhibition induces senescence, apoptosis, and aberrant alternative splicing.** Genetic suppression of PRMT5 has been

reported to impair cancer cell growth by inducing senescence and apoptosis[10,12]. Indeed, we observed an accumulation of enlarged, flattened cells, characteristic of cellular senescence, in GSC lines upon PRMT5 inhibition. We investigated these phenotypic responses using four representative GSC lines, G411, G583, G729, and G797. After 5 days of treatment, we observed a significant

**Fig. 2 GSC lines show differential sensitivities to PRMT5 inhibition. a** Area above the curve (AAC) calculated from dose–response assays across 46 patient-derived GSC lines for GSK591 (red) and LLY-283 (blue) over a range of compound concentrations from 3 nM to 30 μM. DMSO was used as a control. A higher AAC represents greater sensitivity. Asterisk (*) denotes lines studied in more detail in this paper. **b** Common genomic alterations (derived from whole-genome sequencing data), found in 46 patient-derived GSC lines alongside the TCGA classification for each line (in the order of decreasing GSEA enrichment). GSC lines are ordered as in **a**. **c** Western blots for MTAP, CDKN2A, and GAPDH expression across the panel of GSC lines (ordered by increasing response to PRMT5 inhibitors and roughly aligned with **b**). The full GSC line panel experiment was run once, albeit a subset of GSC lines were run twice with similar results. **d** Quantification of senescence-associated β-galactosidase-positive cells in four representative GSC lines. Cells were treated with 1 μM of the PRMT5 inhibitors, GSK591 and LLY-283, and controls, SGC2096 and DMSO, for 5 days. $n = 3$, mean ± SD. Two-tailed unpaired $t$ test with Welch's correction. $p$ values: G411-GSK/SGC = 0.0079, G411-LLY/DMSO = 0.0035; G583-GSK/SGC = 0.1052, G583-LLY/DMSO = 0.0183; G729-GSK/SGC = 0.0071, G729-LLY/DMSO = 0.0076; G797-GSK/SGC = 0.0735, G797-LLY/DMSO = 0.0216. **e** Quantification of Annexin V+ cells in four GSC lines treated with 1 μM of the PRMT5 inhibitors, GSK591 and LLY-283, and controls, SGC2096 and DMSO, for 9–12 days (until cells of DMSO control were confluent). Data shown are representative of two independent experiments. **f** Cell numbers in GSC lines, G561, G583, G837, and G411 treated for 14 days with PRMT5 inhibitors, GSK591 and LLY283, after which drug was either washed out or left on and followed for another 14 days. Dashed line depicts the 14-day point after which the drug was washed out. Data shown are representative of two independent experiments. *$p < 0.05$; **$p < 0.01$. See also Supplementary Fig. 2. Source data are provided as a Source data file.

increase in the percentage of cells that stained positive for senescence-associated β-galactosidase (SA-β-gal) in all four GSC lines (Fig. 2d). We also observed a significant increase in the proportion of cells that stained positive for the apoptotic marker, Annexin V+ (Fig. 2e). Both these effects were more pronounced in LLY-283-treated samples, consistent with the fact that LLY-283 was more potent than GSK591 in attenuating GSC growth. To determine longer-term effects of PRMT5 inhibition on GSC growth, we treated four of our PRMT5 inhibitor-sensitive GSC cell lines, G411, G561, G583, and G837, with 1 μM LLY-283 or GSK591 for 14 days, after which the inhibitors were washed out (see "Methods"). LLY-283 was effective at suppressing growth in 3 out of the 4 GSC lines for up to 14 days after the compound was removed (Fig. 2f).

One of the three GSK591-treated GSC lines (G583) also remained growth suppressed 14 days after treated compound wash-out, while the other three GSK591-treated lines resumed growth within a few days after the compound was removed (Fig. 2f). These data suggest a long-term effect of PRMT5 inhibition that can last beyond direct administration of the compound across multiple patient sample genotypes, a promising attribute for therapeutic scenarios.

To better understand the cellular mechanisms that may account for the attenuation of growth observed upon PRMT5 inhibition in GSCs, we performed bulk RNA-seq on three GSC lines (G561, G564, and G583) after a 3-day treatment with GSK591 or the inactive control, SGC2096, and analyzed the differential effect on gene expression. We identified 646 genes that were significantly differentially expressed between the two treatments (false discovery rate (FDR) <0.05) (Fig. 3a). Gene set enrichment analysis (GSEA) of the differentially expressed genes (DEGs) revealed enrichment in genes involved in spliceosome complex-related pathways, corroborating previous reports that PRMT5 inhibition has a profound effect on the maintenance of splicing fidelity through disruption of arginine methylation of splicing factors and RNA-binding proteins[29]. We also observed enrichment of genes in pathways associated with apoptosis and downregulation of genes involved in G1/S transition, consistent with the increase in SA-β-gal and apoptotic cells observed upon PRMT5 inhibition.

RNA-seq data from the above three GSC lines were analyzed to identify disruptive alternative splicing events (ASEs). A total of 11,582 statistically significant differentially spliced events were identified in the three GSC lines upon PRMT5 inhibition. Only a fraction of these events (3%, 317 events) were common in all three GSC lines, suggesting that, although PRMT5 inhibition leads to widespread splicing disruption in our GSC lines, the alternatively spliced transcripts varied widely from sample to

sample (Fig. 3c). The 317 common ASEs (occurring in 274 genes) included cassette exons (CEs), mutually exclusive exons (MXEs), alternative splicing at the 3' or 5' site (A3/5SSs), and retained introns (RIs), with the highest number of ASEs comprising CEs and RIs (Supplementary Fig. 3a, detailed annotation in Supplementary Data 1). Analysis of the predicted protein impact of the ASEs by VASTdb[35] revealed that 69% of ASEs had predicted disruptive effects on the proteins they encode, with CEs and RIs resulting in the most deleterious impact on their targets (Fig. 3d). Relative to the null expectation whereby each ASE should be equally likely to be included or excluded after treatment, independently of its predicted effect, ASEs observed after PRMT5 inhibition were significantly more often predicted to cause open reading frame (ORF) disruption ($p = 3.9e-64$, Fisher's exact test). Moreover, ASEs predicted to disrupt the ORF upon inclusion were much more likely to be preferentially included ($\Delta PSI > 0$) in PRMT5i-treated cells, whereas ASEs predicted to disrupt the ORF upon exclusion were much more likely to be excluded ($\Delta PSI < 0$) than included in PRMT5i-treated cells. Transcripts affected by disruptive ASEs were enriched in pathways associated with cell cycle progression, correlating with the increase in apoptosis and cellular senescence observed in GSC lines upon PRMT5 inhibition (Supplementary Fig. 3b).

To confirm whether the ASEs indeed affected protein levels, we performed proteomic analysis on the same three GSC lines G561, G564 and G583 treated with 1 μM either GSK591, LLY283, or the inactive control, SGC2096. Overall, the expression differences (fold change observed after PRMT5 inhibition) as measured by RNA-seq and proteomics were weakly but significantly correlated ($r(5004) = 0.154$, $p = 2.2e-16$). The correlation was greater when only proteins whose genes were affected by an ASE in at least one GSC line were considered ($r(936) = 0.2324$, $p = 5.7e-13$) (Supplementary Data 2).

As predicted, proteins whose genes were affected by ASEs were enriched among the most downregulated proteins after PRMT5 inhibition (Fig. 3e). From the 194 proteins detected from the list of genes with ASEs preferentially included after PRMT5i ($\Delta PSI > 0$), we observed a 6.6-fold enrichment ($p = 1.3e-7$, Fisher's exact test) and 2.5-fold enrichment ($p = 6.4e-10$) for the top 50 and top 500 most under-expressed proteins (Fig. 3e and Supplementary Data 3). Similarly, ASEs preferentially excluded after PRMT5 inhibition ($\Delta PSI < 0$) were also predictive of lower protein levels. From the 127 proteins detected, we observed a 7.6-fold enrichment ($p = 1.7e-6$) and 2.6-fold enrichment ($p = 3.9e-9$) among the top 50 and top 500 most downregulated proteins after PRMT5i. This tendency was even more evident when only events predicted to disrupt the ORF were considered. Proteins encoded by genes affected by disruptive ASEs

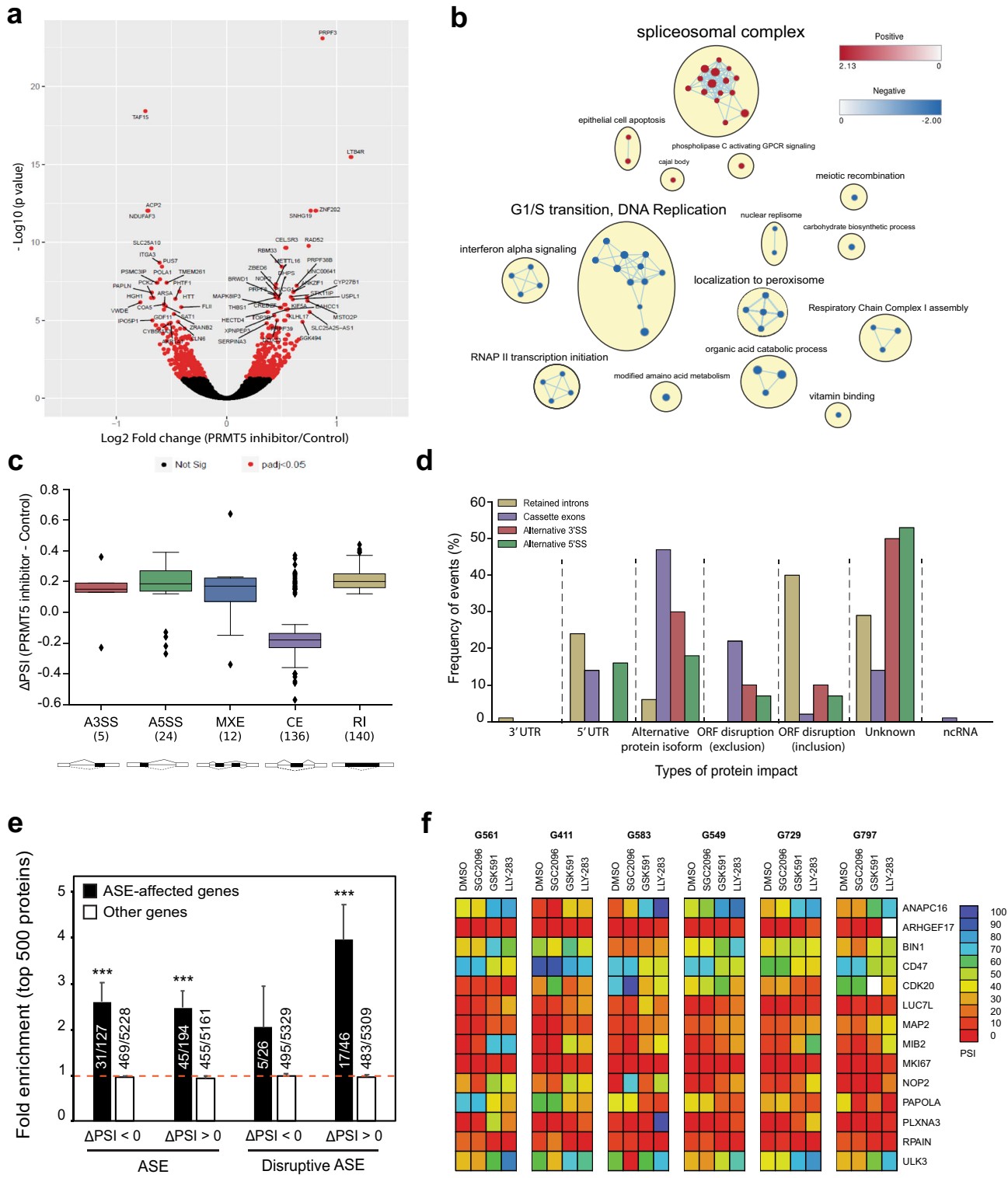

preferentially included after PRMT5 inhibition ($\Delta PSI > 0$) exhibited a 11.6-fold enrichment ($p = 5.9e-5$) and 4-fold enrichment ($p = 3.2e-7$) among the 50 and 500 most depleted proteins (Fig. 3e and Supplementary Data 3).

Taken together, these data uncover a deep effect of PRMT5 inhibition on the proteome that translates into global protein depletion likely mediated by nonsense-mediated decay.

To further investigate these ASEs, we focused on the genes with CEs and RIs, since these were predicted to be the most deleterious on protein function. Fourteen of these genes were annotated to be related to cell cycle. We designed primers for reverse transcriptase polymerase chain reaction (RT-PCR) validation of these ASEs using RNA extracted from six representative GSC lines, including three good responders (G583, G561, and G564) and three poor

**Fig. 3 PRMT5 inhibition leads to deregulation of alternative splicing, affecting regulators of cell cycle. a** Volcano plot comparing fold change (*x*-axis) and *p* value obtained from DESeq2 analysis (*y*-axis) of the expressed genes between GSC lines treated with GSK591 and SGC2096 (*n* = 3). Red dots indicate significantly differentially expressed genes. The top 60 differentially expressed genes are annotated. **b** Gene-set enrichment analysis (GSEA) of all ranked differentially expressed genes (DEGs), visualized in Cytoscape. Networks of related ontologies (shown as colored nodes (red—upregulated; blue—downregulated) connected by blue lines, representing common genes between gene sets are circled and have been assigned group labels. **c** Distribution of ΔPSI (difference in "percent spliced in") for classes of alternative splicing events (ASEs). The box shows the quartiles of the dataset while the whiskers extend to show the rest of the distribution, except for points that are determined to be "outliers." The median marks the mid-point of the data and is shown by the line that divides the box into two parts. Twenty-five percent of scores fall below the lower quartile value while 75% of the scores fall below the upper quartile value. Thus 25% of data are above this value. The box plot shows the middle 50% of scores (i.e., the range between the 25th (Q1) and 75th (Q3) percentile). The minimum is the lowest score, excluding outliers (shown at the bottom of the lower whisker). The maximum is the highest score, excluding outliers (shown at the top of the upper whisker). The upper and lower whiskers represent scores outside the middle 50% (i.e., the lower 25% of scores and the upper 25% of scores) that are not outliers. Values that are lower than the minimum score or higher than the maximum score are considered as outliers. The *y*-axis shows the ΔPSI values between PRMT5 inhibitor and control and the *x*-axis indicates the type of ASEs: Alternative 5′ and 3′ splice sites (A3/A5SS; red and green respectively), mutually exclusive exons (MXE; blue), cassette exons (CE; purple), retained introns (RI; beige). **d** Prediction of protein impact (PI) classes for ASE events. The *y*-axis represents the frequency of occurrence for each type of protein impact prediction classes. The *x*-axis shows the type of PI classes found and their associated type of ASE: A3SS, A5SS, CE, RI. **e** The presence of genes affected by alternative splicing was evaluated among the top 500 enriched or depleted proteins identified by proteomics after PRMT5 inhibition in the same 3 GSC lines (G561, G564, and G583) used for RNA-Seq. Graph represents fold enrichment over expectation considering the total number of detected proteins for each category. ASE: includes genes with alternative splicing events detected in at least 2/3 GSC lines. Disruptive ASE: alternative splicing event predicted to disrupt the open reading frame in at least 2/3 PRMT5i-treated GSC lines. Error bars represent the standard error. ***p* value < 0.001. **f** Heatmap of percent spliced in (PSI) values derived from RT-PCR analysis for 14 PRMT5-dependent ASEs in samples from 6 GSC lines treated with 1 μM GSK591, LLY-283, SGC2096 (inactive control), or vehicle for 3 days. End-point RT-PCR reactions were analyzed via capillary electrophoresis and percent spliced in (PSI) values were calculated as described in the "Methods" and plotted as a heatmap. Source data are provided as a Source data file.

responders (G729, G797, and G549) to PRMT5 probes treated with GSK591, LLY-283, or inactive control. Semi-quantitative RT-PCR followed by capillary electrophoresis revealed significant shifts in splice isoform levels (as predicted by the RNA-seq analysis) in the PRMT5 probe-treated samples for 75% of these ASEs (Fig. 3f and Supplementary Data 4). The concentrations of each splice variant were annotated as previously described, and the relative abundance of variants was expressed as a percent splicing index (psi or Ψ) calculated as the percentage of the amplicon concentration of the longest variant relative to the total amplicon concentration of both long and short variants[36,37]. We observed significant shifts in percent spliced-in (PSI) values in GSC samples treated with either PRMT5 inhibitor for the ASEs in *ANAPC16*, *BIN1*, *CD47*, *CDK20*, *MAP2*, *MIB2*, *RPAIN*, *NOP2*, *LUC7L*, *PAPOLA*, *PLXNA3*, and *ULK3* genes (summarized as a heatmap in Fig. 3f, see also Supplementary Fig. 4 for capillary gel images and Supplementary Data 4 for detailed peak annotation and PSI data). The PSI shifts observed with the two different PRMT5 inhibitors were in the same direction in all cases (Fig. 3f, Supplemental Fig. 4, and Supplementary Data 4) and conserved across all six GSC lines tested, thus validating the RNA-seq data.

A number of the ASEs are associated with protein isoforms that have been shown to negatively affect proliferative or cell cycle programs. For example, the *BIN1* alternative exon 12A has been shown to inhibit the Myc–BIN1 interaction and thus deactivate the tumor-suppressor function of the BIN1 protein[38,39]. Another example is the *MKI67* exon 7 inclusion in the KI67 cell proliferation antigen, a well-known cell proliferation marker[40]. The long isoform with inclusion of this exon has been reported to inhibit cell proliferation[41]. Similarly, skipping of the *CDK20* exon we identified is reported to result in a frameshift and a protein with a different C-terminal half that is unable to interact with cyclins and phosphorylate CDK2 thus impairing its cell cycle function[42]. Moreover, depletion of the full-length CDK20 isoform impairs GBM cell proliferation[43,44].

Moreover, half of the PRMT5 inhibitor-induced splicing changes are predicted to disrupt the ORF of the encoded proteins and thus result in a loss of function of the respective gene. This is the case for *RPAIN*, *CDK20*, *MIB2*, *NOP2*, *ULK3*, *ARHGEF17*, and *PLXNA3* splice variants. For example, alternative splicing of

three consecutive exons in the *RPAIN* gene (encoding for the RPA interacting protein) mapping in the C-terminal Zn-finger domain, which we found to be preferentially skipped upon PRMT5 inhibition, leads to truncated proteins missing all or parts of this domain[45], which in turn alters their subcellular localization (nuclear vs cytoplasmic for the canonical *RPAIN* isoform) and function in the import of RPA in PML nuclear bodies[46] and impairing cell proliferation[47]. This splicing-mediated alteration in protein function of key nuclear, cell cycle proteins may explain a strikingly abnormal nuclear morphology observed in our GSC lines upon PRMT5 chemical probe treatment (Fig. 4). In multiple GSC lines, we find large multi-lobed nuclei resembling donuts in the following treatment with either PRMT5 inhibitor. Interestingly, this phenotype has also been observed upon silencing of *ARHGEF17*, *RPAIN*, and *ANAPC16*, which undergo preferential intron retention (*ARHGEF17*), or exon skipping for the latter two genes, upon PRMT5 inhibition, has been shown to induce chromosome defects during mitosis, leading to binucleation and poly-lobed nuclei[48]. Furthermore, similar nuclear morphology changes have been observed during normal granulocyte differentiation and are functionally linked to an orchestrated intron retention program[49]. Taken together, these data identify a cohort of altered splicing events caused by PRMT5 inhibition resulting in strong phenotypic effects on cell cycle and proliferation as well as nuclear morphology.

These strong effects on splicing led us to investigate whether inherent or pre-existing differences in alternative splicing may account for the differential sensitivities to PRMT5 inhibition observed among GSC lines. We extracted ASEs from RNA-seq data of 31 representative GSC lines at baseline and compared their inclusion levels between lines that respond well to PRMT5 inhibitors ("good responders") to "poor responders" (as defined by AAC—see Fig. 2). The top 45 differential ASEs that distinguish these two groups are summarized in Fig. 5a and detailed in Supplementary Data 1. We observed 29 ASEs with higher inclusion levels (dPSI > 0) in the good responders, in contrast with 16 exclusion events (dPSI < 0) which have higher inclusion in the poor responders. Gene annotation and enrichment analysis revealed significant enrichment in gene products involved in

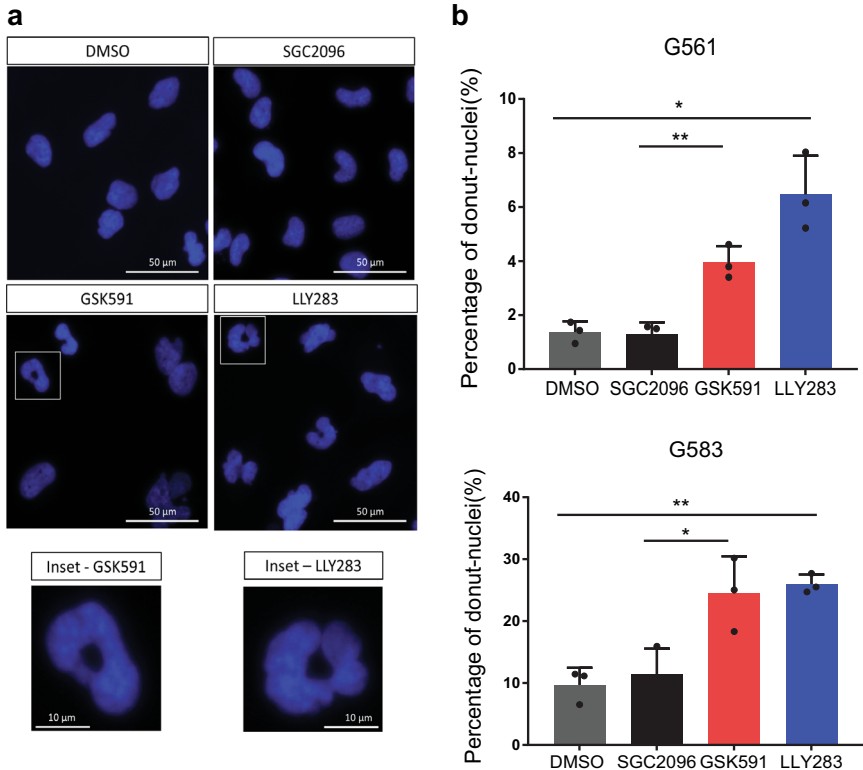

**Fig. 4 Effect of PRMT5 probes on nuclear morphology. a** Hoechst staining of the GSC line, G561, upon treatment with the PRMT5 inhibitors, GSK591 and LLY-283, at 1 μM, as well as the control compounds, DMSO and SGC2096. Scale bar, 50 μm. Inset scale bar, 10 μm. **b** Percentage of cells with donut-shaped nuclei in the GSC lines, G561 and G583, 5 days after treatment with 1 μM of the PRMT5 inhibitors, GSK591 and LLY-283, and controls, SGC2096 and DMSO. $n = 3$ biologically independent samples, mean ± SD. Two-tailed unpaired $t$ test with Welch's correction. $p$ values: G561-GSK591/SGC2096 = 0.0050, G561-LLY-283/DMSO = 0.0191, G583-GSK591/SGC2096 = 0.0406, G583-LLY-283/DMSO = 0.0025. *$p < 0.05$; **$p < 0.01$. Source data are provided as a Source data file.

cytoskeletal processes among these ASEs (Fig. 5b). This data suggests that this group of patient-derived cells displays inherent differences in specific splicing patterns, which can distinguish their response to PRMT5 inhibition.

**LLY-283 crosses the blood–brain barrier (BBB) and significantly extends survival of mice with orthotopic GBM xenografts.** To evaluate whether PRMT5 has potential as a therapeutic target in GBM, we investigated the ability of the chemical probe LLY-283, the more potent PRMT5 inhibitor, to cross the BBB. We administered 25, 50, and 100 mg/kg of LLY-283 to mice through oral gavage and collected both plasma and brain samples after 24 h. We detected LLY-283 concentrations of 159 nM in the plasma and 269 nM in the brain after 24 h of dosing with 100 mg/kg, suggesting that the compound clears out of the plasma more quickly than the brain (Fig. 6a). The concentration of LLY-283 in the brain at a dose of 50 mg/kg (119 nM) exceeds the cellular IC90 value for growth suppression of G411 while showing significantly reduced concentrations in the plasma as compared to that with 100 mg/kg, making it a reasonable dose for our in vivo study.

We further performed tolerability studies to select a dosage regimen that can be tolerated by the NOD SCID gamma (NSG) mice. LLY-283 was found to induce a significant (>20%) loss in body weight of mice when administered every other day over 3 weeks (Supplementary Fig. 5a). However, when administered consecutively for 3 days followed by no dosing for 4 subsequent days (3 days on, 4 days off) in weekly cycles, LLY-283 was tolerated for up to 8 weeks, albeit with moderate body weight loss (<10%; Supplementary Fig. 5b).

To test the potential therapeutic efficacy of PRMT5 inhibition in an intracranial xenograft model of GBM, we transplanted 20,000 GSCs (G411) into NSG mice ($n = 24$) and initiated treatment 1 week after intracranial injection. Mice were treated with either vehicle ($n = 12$) or 50 mg/kg LLY-283 ($n = 12$) in weekly cycles of 3 days on, 4 days off, until they were sacrificed (Supplementary Fig. 5c). Two mice from the vehicle group were censored due to aberrant tumor growth outside the skull, attributed to technical complications.

Mice treated with LLY-283 showed significant extension of life span compared to the vehicle-treated mice (median survival; LLY-283 = 37 days, vehicle = 30 days; $p = 0.0046$; Fig. 6b). Hematoxylin and eosin (H&E) staining of brain sections showed that vehicle-treated mice displayed large tumors with relatively discrete borders characterized by areas of pseudopalisading necrosis, distinctive of GBM, while LLY-283-treated mice showed relatively small areas of dense cell growth, potentially explaining the increased survival of mice in this group (Fig. 6c). Importantly, the SDMA mark was significantly reduced in the xenograft tissues of mice treated with LLY-283 as compared to the vehicle-treated mice (Fig. 6d), suggesting that the reduction of tumor size and concomitant increase in survival are due to the disruption of the enzymatic function of PRMT5. Taken together, these data demonstrate that PRMT5 is a promising therapeutic target for the treatment of GBM and provide a strong basis for the clinical development of a brain-penetrant PRMT5 inhibitor.

## Discussion
The identification of new therapeutic options for primary GBM has been confounded by intra- and inter-tumoral heterogeneity, a

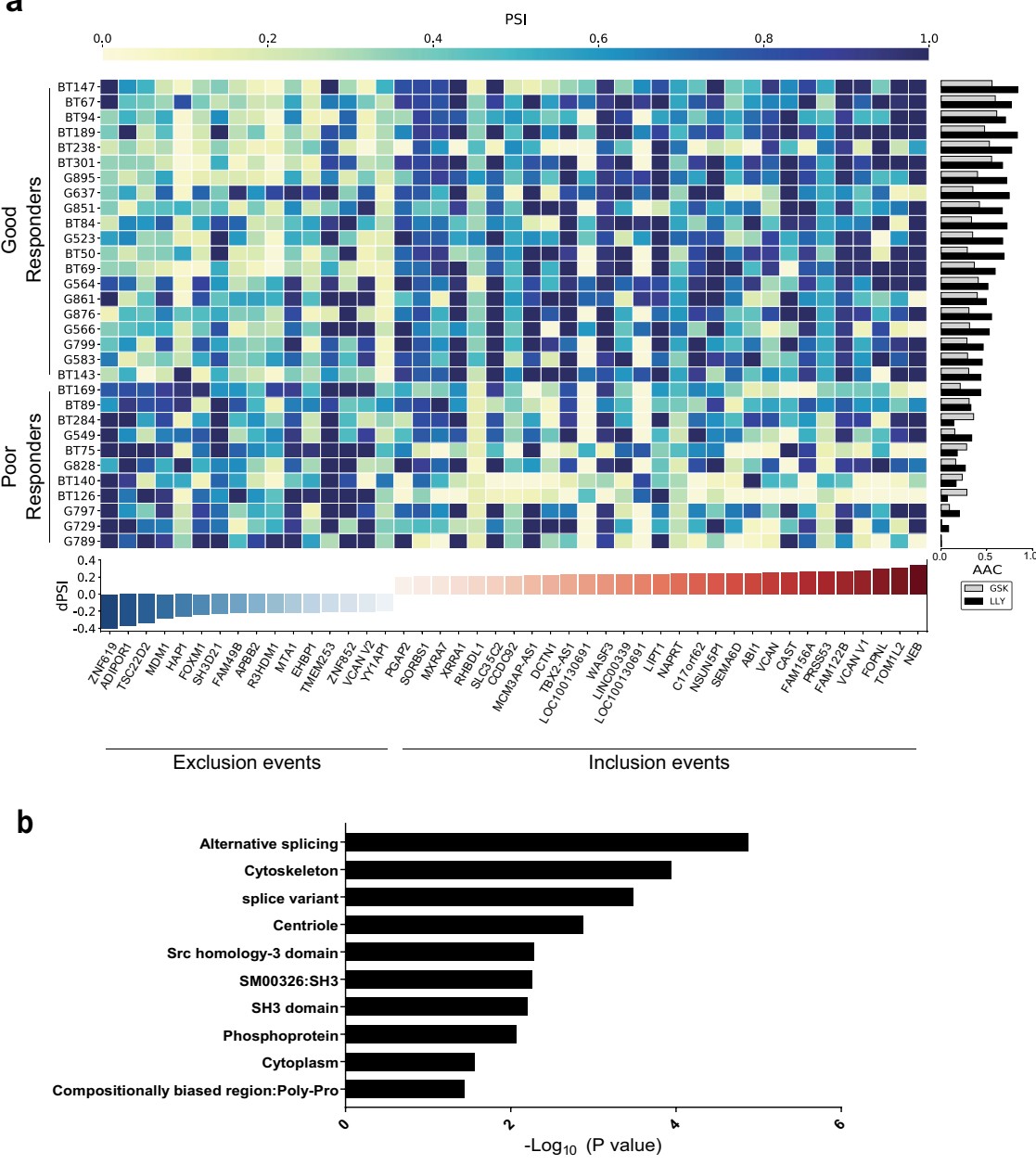

**Fig. 5 A splicing signature predictive of PRMT5 inhibitor response. a** Heatmap of percent spliced in (PSI) values for 45 differential ASEs in transcripts across 31 GSC lines that are either good (19 lines) or poor responders (12 lines) to PRMT5 inhibition alongside AAC. More details about each ASE are provided in Supplementary Data 1. **b** Significant gene ontology terms and cellular processes derived from GO enrichment analysis of the 45 predictive ASEs. Enrichment analysis for transcripts containing predictive alternative splicing events for Gene Ontology (GO) Biological Process. The *y*-axis shows the top 10 enriched classes, ranked by adjusted *p* value (one sided, Fisher's exact test) from low to high. The *x*-axis shows the *p* value transformed in a −log10 scale for visualization purposes. The *p* values were (from top to bottom in the chart order): 1.34E−05, 1.13E−04, 3.26E−04, 0.001315, 0.005137, 0.005137, 0.005507, 0.006227, 0.008603, 0.027258. Source data are provided as a Source data file.

paucity of BBB permeable drugs, and the presence of a drug-resistant tumor-initiating stem cell population[4,5,23,50]. The absence of truncal genetic mutations in these tumors has made targeted therapies challenging and suggests that disruption of more widespread cellular processes is necessary. In this study, we demonstrate that PRMT5 inhibition using either of two potent and selective small molecule chemical probes is effective in attenuating the growth of a large cohort of GSC lines derived from adult and pediatric GBM patients. We also show that PRMT5 inhibition significantly reduces the SFC, a hallmark of CSCs, of both GSC lines and primary, never before cultured GBM

patient cells. These results suggest that PRMT5 inhibition may be effective at targeting the population of GBM cells that drive disease progression and promote resistance to radiotherapy and chemotherapy[3,4], which will be required for long-term disease control. Importantly, PRMT5 inhibition was effective across a highly diverse cohort of patient-derived GSCs with distinct genotypes. This result is consistent with results from a CRISPR-Cas9 genome-wide screen, which demonstrated PRMT5 as an essential gene across 8/8 patient-derived GSCs[51]. Interestingly, GSCs that were more sensitive to PRMT5 inhibitors were more likely to be classified as proneural based on transcriptomics data, suggesting

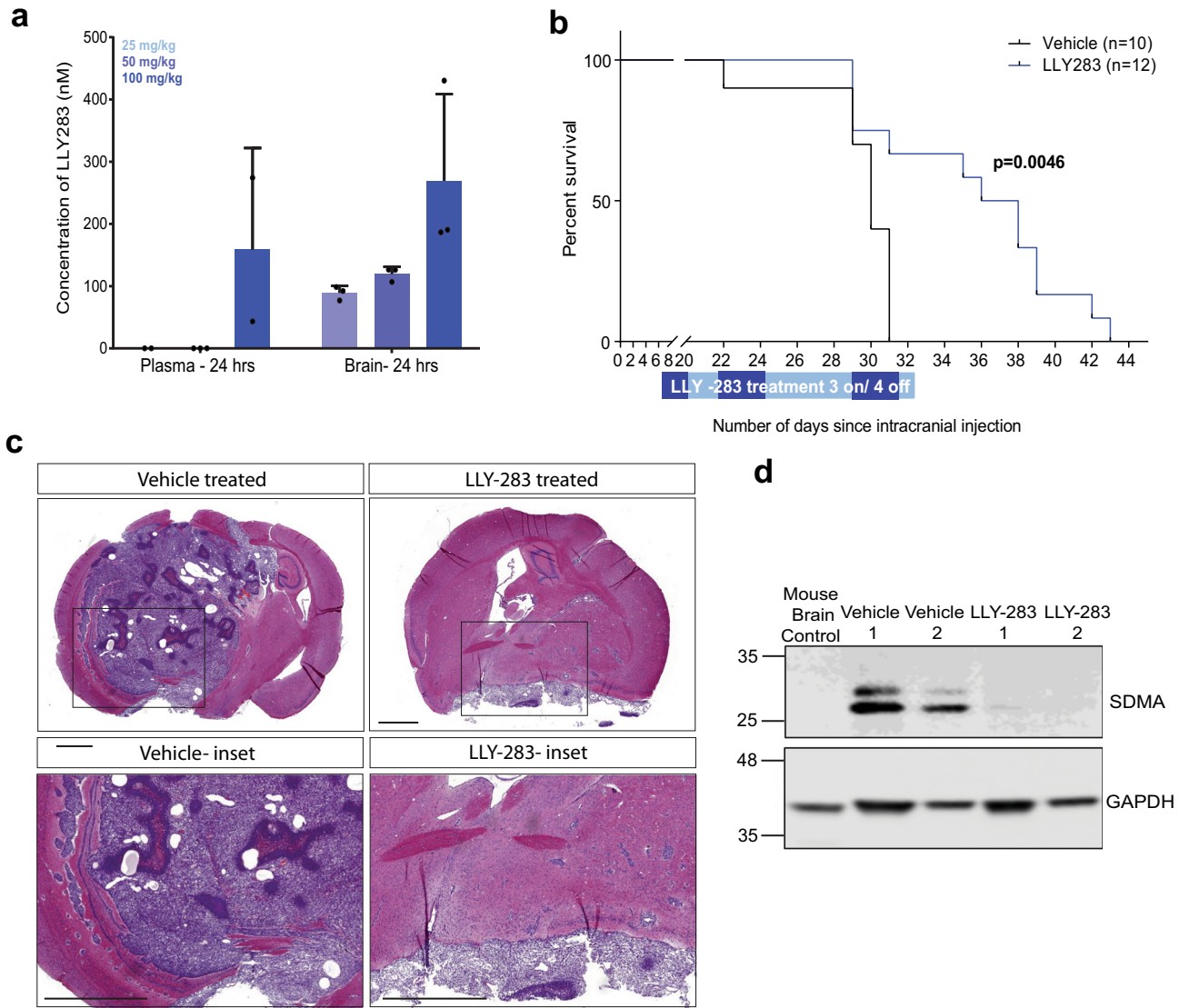

**Fig. 6 Pharmacological inhibition of PRMT5 significantly extends survival of mice with orthotopic xenografts of GSCs. a** Concentration of LLY-283 in the plasma and brain of NSG mice 24 h after oral administration of 25, 50, and 100 mg/kg of LLY-283. $n = 3$, mean ± SD. **b** Kaplan–Meier survival curve of immunodeficient NSG mice injected intracranially with G411 GSCs and treated with 50 mg/kg LLY-283 (Vehicle: $n = 10$; LLY-283: $n = 12$). Significance was estimated using the log-rank (Mantel–Cox) test (two sided). Chi square = 8.038, $p$ value = 0.0046. **c** H&E staining of a mouse brain with orthotopic xenografts treated with LLY-283 or vehicle at end point (scale bar = 1000 μm). $N = 5$ for LLY-283, $N = 2$ for vehicle. **d** Representative western blot of symmetric dimethyl arginine mark in tumor tissue from mice with orthotopic xenografts treated with LLY-283 or vehicle at experimental end point. The experiment was repeated twice with similar results. See also Supplementary Fig. 5. Source data are provided as a Source data file.

that patients with the proneural subtype of GBM may be more likely to derive benefit from treatment with PRMT5 inhibitors.

PRMT5 has been shown to play an important role in the maintenance of splicing fidelity through the methylation of Sm proteins, which is critical to the formation of the mature small nuclear ribonucleoprotein complex[29,52]. We showed that PRMT5 inhibition in patient-derived GSC lines led to significant changes in gene expression and had a profound effect on the spliceosome pathways, leading to increased disruption in alternative splicing, particularly in pathways related to cell cycle progression. Moreover, we showed that the PRMT5i-induced splicing changes led to a disruptive proteome with significant protein downregulation among the miss-spliced genes. Given that PRMT5 has multiple protein targets, we postulate that the ASEs affected by PRMT5 inhibition perturb diffuse cellular processes governing individual GBM lines, resulting in a spectrum of phenotypic effects. It is worthwhile to note that most of these ASEs were also found in

other cancer cell lines following PRMT5 inhibition[28,34] indicating that these are bona fide consequences of PRMT5 inhibition. Moreover, the same *MKI67* exon was also found to shift with a Type I PRMT inhibitor[53]. Furthermore, while we identify events that are common to all the GSC lines assayed, it must be noted that there were thousands of other ASEs that were unique to each patient sample, which may also contribute to variable sensitivities observed across our samples.

Given that patient stratification is one of the major challenges in translating preclinical results into successful clinical outcomes, identification of a predictive biomarker of sensitivity to PRMT5 inhibition may enable stratification of patients who are more likely to respond to therapy. Interestingly, while previous studies have linked the sensitivity of cancer cell lines to PRMT5 inhibition to *MTAP* status[30], *TP53* mutations[28], or *CLNS1A/RIOK1* ratio[34], none of these factors correlated with response in our patient-derived samples. On the other hand, a set of 45

pre-existing ASEs were differentially spliced in the more sensitive GSC lines as compared to poor responders GSC lines (Fig. 5), showing promise as potential biomarkers of response. Notably, several of the encoded protein isoforms like ABI1, DCTN1, EHBP1, HAP1, FOPNL, and WASF3 are involved in cytoskeletal dynamics such as actin and microtubule binding as well as centrosome binding and ciliogenesis, which are essential processes for GSC division, proliferation, migration, and self-renewal. Moreover, another subset composed of APPB2, CAST, and HAP1 proteins are linked to neurodegeneration and are involved in APP processing while the ABI1 and APPB2 splice variants are previously described cancer-associated splice switches regulated by the neuronal splicing factor RBFOX2[37]. Intriguingly, two alternative isoforms of the same gene, *VCAN*, encoding for the extracellular matrix proteoglycan Versican were found among this set of splices: the V1 isoform, which includes the exon encoding for the GAGα repeats that we found to be enriched in the good responders is known to promote neuronal differentiation and cell proliferation[54,55]. In contrast, the Versican V2 isoform, encompassing the exon encoding for the GAGβ repeats, was preferentially included in the poor responders (Fig. 5) and inhibits axonal growth and cell proliferation[54,55]. This suggests that there are inherent differences in neuronal differentiation potential and possibly proliferation as well as drug response capacity in the GSC lines that respond well to PRMT5 inhibitors from those that do not respond well. This is entirely consistent with our previous observations on neuronal differentiation capacity in a subset of GSC lines[26] and our current observation that the good responders are enriched in the proneural TCGA subgroup of GSC samples. Finally, poor responder GSC cell lines harbored a splice variant of the TSC22D2, a protein that has been reported to interact with WDR77/MEP50, a subunit of the PRMT5 enzymatic complex[56]. This suggests a potential functional link with PRMT5.

Many new drugs and therapies for GBM fail in clinical trials because they do not effectively cross the BBB. We were able to demonstrate that LLY-283, the more potent of the two PRMT5 inhibitors, was able to efficiently cross the BBB; this is the first report of a brain-penetrant PRMT5 inhibitor. Furthermore, when administered orally, LLY-283 conferred significant survival benefit in mice with orthotopic patient-derived GBM xenografts, a preclinical model of GBM, although more preclinical studies with clinical compounds are warranted in future study.

Our work reveals important insights that link PRMT5 inhibition to the disruption of constitutive splicing, impairing both proliferation and self-renewal in GBM CSCs and providing a strong rationale for targeting PRMT5 in GBM. Our proof-of-concept in vivo study using brain-penetrant chemical probe suggests that a brain-penetrant PRMT5 drug may be an effective therapeutic strategy. Considering that GBM patients that lack MGMT promoter methylation derive minimal therapeutic benefit from temozolomide treatment[57], PRMT5 inhibition instead of temozolomide, in combination with established surgical and radiotherapy protocols, may improve outcomes for this cohort of patients. Finally, our data on pediatric GSCs also suggests that PRMT5 is an exploitable vulnerability in pediatric GBM, for which no effective chemotherapeutic options exist.

## Methods

**Patient samples and patient-derived cell lines.** Fresh tumor samples were obtained from patients during their operative procedure following informed consent. All experimental procedures were performed in accordance with the Research Ethics Board at The Hospital for Sick Children (Toronto, Canada) and University of Toronto (Toronto, Canada). All primary GBM tissues were obtained from The Hospital for Sick Children (Toronto, Canada), Toronto Western Hospital (Toronto, Canada), St. Michael's Hospital (Toronto, Canada), or Foothills Medical Centre, University of Calgary (Calgary, Canada). All primary human fetal (HF)

samples were obtained from Research Centre for Women's and Infants' Health BioBank (Toronto, Canada) and Mount Sinai Hospital (Toronto, Canada). GSCs and HF samples were derived as previously described[25].

Each sample ID follows the structure GXXX, BTXX, or HFXXX, where XX(X) is the numerical identifier of the primary sample. G: Glioblastoma neural stem cell lines grown adherently; BT: Glioblastoma neural stem cell lines grown as freefloating spheres; HF: Human fetal neural stem (HFNS) cell lines. The suffix "r" denotes a recurrent tumor and the suffix "p" denotes tumors obtained from pediatric patients.

GXXX and HFXXX samples were grown adherently in serum-free medium as described previously[26]. Briefly, cells were grown on PRIMARIA™ culture plates (Corning) coated with poly-L-ornithine (Sigma) and laminin (Sigma) and maintained in Neurocult NS-A basal medium (human) (StemCell Technologies) containing 2 mM L-glutamine (Wisent), 75 μg/mL bovine serum albumin (Life Technologies), in-house hormone mix equivalent to N2 (home-made), B27 supplement (Life Technologies), 10 ng/mL recombinant human epidermal growth factor (rhEGF; Sigma), 10 ng/mL basic fibroblast growth factor (bFGF; StemCell Technologies), and 2 μg/mL heparin (Sigma). Cells were passaged using enzymatic dissociation with Accutase (StemCell Technologies).

For GSCs grown in suspension (BTXX lines), cells were cultured as free-floating spheres in serum-free media as previously described[58]. Briefly, GSCs were maintained in serum-free media supplemented with EGF and bFGF (20 ng/mL each, Peprotech) and 2 μg/mL heparin sulfate (Sigma). Upon reaching 150–200 μm, spheres were dissociated via mechanical trituration or with Accumax (Innovative Cell Technologies, Inc.) and re-plated as single-cell suspensions in T25 flasks.

**Animal studies.** All animal-related experimental procedures were approved by The Hospital for Sick Children's Animal Care Committee. Five-to-10-week-old NSG female mice were used for in vivo studies. Mice were housed at The Hospital for Sick Children Laboratory Animal Services. Sample sizes for experiments were determined based on previous experience, without power calculations.

**Epigenetic probe library.** We used 39 well-characterized chemical probes from the SGC. Each compound selectively and potently inhibits a specific epigenetic protein and has significant cellular activity at ≤1 μM. The list of epigenetic probes along with their protein targets and relevant publication describing their properties is in Supplementary Table 1.

**Epigenetic probe screen.** Two thousand GSC, BT, or HF cells were plated adherently in 384-well CELLBIND™ plates (Corning) and imaged using the IncuCyte® ZOOM Live-Cell Analysis System (Essen Biosciences). Cells were imaged with a ×10 objective using phase-contrast every 4–8 hours (depending on the experiment) until the experimental end point. Culture media was refreshed every 5 days with ×1 chemical probe concentration. Cell confluency or fluorescence was analyzed using the IncuCyte Live Cell Analysis System (Essen Biosciences), in which an algorithm was used to approximate the cell confluency of each well. The processing definition was tailored for each cell type, allowing for reproducible and robust monitoring of cell confluency over time. The data represented in the heatmap is $\log_2$ of the average confluency of triplicates in the probe wells normalized against the average confluency in the DMSO control wells.

**Cell proliferation assay.** GSCs were plated in triplicate in 96-well PRIMARIA™ plates (Corning). Cells were treated with PRMT5 inhibitors (1 μM of GSK591 or LLY-283) or controls (1 μM of SGC2096 or 0.05% DMSO) for 9–12 days or until the DMSO control wells were confluent. Culture media was refreshed every 5 days with ×1 chemical probe concentration. Cell growth was monitored using the IncuCyte ZOOM™ Live Cell Analysis System. Confluency was calculated from cell images taken every 8 h using the IncuCyte ZOOM software. To obtain relative confluency (to account for differences in cell-plating density), cell confluency at all time points was normalized to the initial confluency value in each well.

**Dose–response assay.** GSCs were plated in 96-well PRIMARIA™ plates, with three technical replicates per dose, and cultured across nine concentrations (ranging from 3 nM to 30 μM) of PRMT5 inhibitors for 9–20 days until the DMSO control wells reached confluence. Culture media was refreshed every 5 days with ×1 chemical probe concentration. The response of adherent cells (GXXX lines) to PRMT5 inhibition was estimated from the confluency data normalized to the DMSO controls, measured using the IncuCyte ZOOM™ Live Cell Analysis System. The response of lines cultured in suspension (BTXX lines) to PRMT5 inhibition (GSK591 and LLY283) was estimated from cell viability normalized to the DMSO controls, assessed by Alamar Blue assay (Thermo Fisher Scientific) using a Gemini EM Fluorescence Microplate Reader (Molecular Devices), after 14 h of incubation with the active ingredient. Cell viability confluence values were normalized to control end point using IncuCyteDRC R package[59]. The PharmacoGx package was then used to generate drug–response measurements for each line, including the AAC (drug dose–response)[60]. We generated the dose–response curves at the 95th percentile of the total time to reach 100% confluency of the DMSO controls to allow for maximum drug effect and to avoid saturation. The raw dose–response data to reproduce the AAC and IC50 can

be obtained from https://github.com/bhklab/PRMT5i_GBM. The PharmacoSet (PSet) object for the dataset is available at https://doi.org/10.6084/m9.figshare.8220434.

**Protein extraction and western blot**. Lysis buffer containing 10 mM Tris-HCl, 0.5% Triton X-100, 150 mM NaCl, 1 mM EDTA, 10 mM MgCl₂, 1% sodium dodecyl sulfate, protease inhibitors, and benzonase was used to harvest total protein for western blot. In all, 15–35 µg of protein is loaded in each lane for SDMA mark in Figs. 1, S1, S2, and 5. In all, 20 µg of total protein were loaded for MTAP and CDKN2A western blots in Fig. 2. Cell lysates were run on NuPAGE 4–12% Bis-Tris protein gels and transferred onto polyvinylidene difluoride membranes. Primary antibodies used were anti-SDMA (Cell Signaling Technologies, #13222, 1:1000), MTAP (Cell Signaling Technologies, #4158, 1:1000), CDKN2A (R&D Systems, AF5779, 1 µg/mL), and anti-GAPDH (EMD Millipore, MAB374, 1:5000). Secondary antibodies used were IRDye® 680RD anti-mouse IgG (LI-COR, 926-68072, 1:5000) and IRDye® 800CW anti-rabbit IgG (LI-COR, 926-32211, 1:5000). Membranes were visualized on an Odyssey® CLx Imaging System (LI-COR). Full, uncropped images of the western bots are shown in the Source data file.

**Limiting dilution assay**. GSCs were plated in serial dilutions on non-adherent 96-well plates with six technical replicates per dilution in the media described above for adherent cells (see "Patient samples and patient-derived cell lines"). LDAs were completed by plating cells in suspension in 1:2 serial dilution. In detail, the following cell numbers (rounded) were seeded per well of a 96-well plate: 2000, 1000, 500, 250, 125, 63, 31, 16, 8, 4 cells per well. Each dilution was done in six technical replicates, all on the same plate. Drugs were added on the first day and each well was replenished with 50 µL of fresh media with 1× drug every week. After 1 and 2 weeks of plating, each well was scored for the presence of spheres. Readout at assay end point was positive or negative for the presence of spheres in each well. The fraction of technical replicates positive for the presence of spheres at each dilution was entered into the extreme limiting dilution analysis (ELDA) software (http://bioinf.wehi.edu.au/software/elda/) to determine the estimated sphere-forming frequency along with upper and lower limits (denoted by the error bars) with a 95% confidence interval. Data were tested for inequality in frequency between multiple groups and for adequacy of the single-hit model using the ELDA software[61].

**Apoptosis assay**. Cells were plated in 96-well plates and treated with PRMT5 inhibitors or control compounds, as for the cell proliferation assay. At the end of the treatment duration, IncuCyte® Annexin V Red Reagent for apoptosis (Essen Bioscience) was added to the medium (1:200 dilution). The reagent contains Annexin V conjugated to a fluorophore. Upon binding of the Annexin V reagent to externalized phosphatidylserine in apoptotic cells, the reagent emits a green or red fluorescent signal, which is detected by fluorescence live-cell imaging using the IncuCyte Live Cell Imaging System (excitation—green: 440–480 nm, red: 565–605 nm; emission—green: 504–544 nm, red: 625–705 nm). Images were captured 6 h after the addition of reagent. The total number of cells and number of Annexin V positive (fluorescent cells) were analyzed using the integrated IncuCyte ZOOM software (Essen Biosciences).

**SA-β-gal staining**. Senescence assay was carried out as previously described by Debacq-Chainiaux et al.[62]. Briefly, cells were fixed with 4% paraformaldehyde for 5 min at room temperature, then incubated with SA-β-gal staining solution overnight or until cells were stained blue. The reaction was terminated by aspirating the staining solution and washing the cells twice with distilled water. Cells were then counterstained with nuclear fast red for 5 min at room temperature, then washed twice with distilled water. Cells were visualized on an EVOS FL Auto 2 Cell Imaging System (Invitrogen). Senescence was quantified by counting the number of senescent cells stained blue by β-galactosidase out of the total number of cells.

**Cell growth and metabolite harvest for metabolomics analysis**. Cultures grown adherently and labeled GXXX or HFXXXX were seeded into 6-well plates as described above until 60% confluence was reached. One day prior to extraction, culture medium was changed to fresh medium. Twenty hours following the media change, extracellular metabolites were sampled by transferring 200 µL of culture supernatant into 800 µL of Methanol (liquid chromatography–mass spectrometry (LC-MS) grade, Fisher). In parallel, a 200-µL sample of fresh medium was also sampled into 800 µL of Methanol to serve as a media control. Media controls and extracellular metabolite samples were stored at −80 °C until analysis.

Four hours prior to intracellular metabolite extraction, culture media was changed again. Intracellular metabolite extraction was performed by aspirating culture media and rapidly covering the cells with 1.2 mL of ice-cold extraction solvent (80% LC-MS-grade Methanol). Cells were scraped off the bottom of the plate and both solvent and cells were transferred to a clean tube. Coating control samples were collected by adding 1.2 mL of extraction solvent to empty poly-L-ornithine (Sigma) and laminin (Sigma) coated plates and then transferring the solvent to clean tubes. All samples and controls were stored at −80 °C until analysis.

Cell lines grown in suspension (BTXXX) as neurospheres were seeded in T-26 flasks and grown for 7 days. One day prior to collection, culture media was

changed to fresh. Twenty hours after the media switch, extracellular metabolites were harvested. One milliliter of spent culture medium was collected and centrifuged at 14,000 × g for 2 min. A 200 µL of the supernatant was transferred into 800 µL of LC-MS-grade Methanol (Fisher). In parallel, 200 µL of fresh media was transferred to 800 µL of Methanol to serve as media control. All extracellular metabolite samples and controls were stored at −80 °C until analysis.

Four hours prior to intracellular metabolite extraction, media was changed again. In order to extract intracellular metabolites, cells from 5 mL of culture were harvested by centrifugation at 800 × g for 7 min. The supernatant was discarded, and cell metabolism was quenched with 1.2 mL ice-cold extraction solvent. Two controls were harvested in parallel including 1.2 mL solvent blank and 5 mL of fresh media control, which was centrifuged in parallel cell cultures. All samples were stored at −80 °C until analysis.

**Sample preparation**. Metabolite samples were cleared of cells and insoluble material by centrifugation for 7 min at 14,000 × g at 4 °C. Supernatants were transferred to a clean tube. Extraction solvent was dried under N₂ gas using a TurboVap connected to a Parker nitrogen generator operated at 99.5% purity. Dry metabolite extracts were reconstituted in high-purity H₂O and vortexed to mix and resuspended for 1 min at room temperature. Cell pellets were assayed for total RNA using the Quant-iT™ RiboGreen™ RNA Assay Kit (ThermoFisher). Cell pellets were reconstituted in 250 µL Elisa buffer (20 mM Tris-HCl pH 7.5, 150 mM NaCl, 1 mM EDTA, 1 mM EGTA, 1% Triton x-100, 1 mM sodium orthovanadate) then diluted 20× in 10 mM TE (pH 7.5). Diluted samples were combined with dye and incubated for 5 min at room temperature. Fluorescence was read using a Victor Wallac plate reader. The reconstitution volume for each cell extract was determined by total RNA measurement such that the volume of dried extract corresponding to 2 ng of total RNA were reconstituted in 15 µL of water. Extracellular metabolite extract, media controls, and process controls were reconstituted in constant volume of water 75 µL per 1 mL of dried extract. All reconstituted samples were combined with equal volume of ¹³C/¹⁵N internal standard mixture composed of 99.9% ¹³C¹⁵N-labeled *Saccharomyces cerevisiae* extract. Following mixture with standard, samples were once again cleared for 7 min at 14,000 × g at 4 °C and transferred to LC-MS polypropylene conical vials (Agilent). Vials were stored at −80 °C until analysis using LC-MS.

**Chromatographic separation**. All metabolomics samples described in this study were analyzed by chromatographic methods essentially as described in Wan et al.[63]. Briefly, tributylamine ion paired LC was carried out using an Agilent 1290 UPLC system with a 0.25 mL/min flow rate and a 2-µL sample loop and an Extend C18 RRHD 1.8 µm, 2.1 × 150 mm column (Agilent).

**Mass spectrometry**. Data were acquired using an Agilent 6550 QToF instrument. Samples were ionized using an Agilent Jet Stream electrospray ionization source operated in negative mode. Gas temperature in the ion source was 150 °C with a flow rate of 14 L/min. Nebulizer pressure was 45 psig; sheath temperature was 325 °C with a gas flow rate of 12 L/min. Voltage for both capillary and nozzle was 2000 V. The funnel DC voltage was −30 V, funnel voltage drops were −100 and −50 V in the high- and low-pressure funnels, respectively. The RF voltages were 110 and 60 V in the high- and low-pressure funnels respectively. Mass spectra were acquired between 50 *m/z* and 1100 *m/z* with a rate of 2 spectra per second. Mass lock mixture was used as described in Wan et al.[63].

**Data analysis**. Quantitation of MTA was carried out by calculating the area under the curve of a chromatographic trace corresponding to the $[M-H]^-$ ion of MTA ($C_{11}H_{15}N_5O_3S = 296.0823$ *m/z* $[M-H]^-$, "light") and its ¹³C/¹⁵N-labeled variant ($[13\,C]_{11}H_{15}[15\,N]_5O_3S = 312.1044$ *m/z* $[M-H]^-$, "heavy"). Ion chromatograms were extracted from raw profile mass spectra using software developed in the Rosebrock laboratory. Extracted mass windows of 75 ppm around the target mass were locally aligned and manually integrated using the ManuallyIntegrateFeatures function, in order to ensure consistent margins for peak integration. The retention time of 10.5 min was determined by analyzing a neat MTA standard (Sigma) using the LC-MS method as described above. To address sample- and instrument-derived variation, the ratio of "light" to "heavy" integrals was calculated and used as a proxy for concentration of MTA. Additional batch effects stemming from samples collected in different calendar years were corrected by median centering the ratios within each year of sample acquisition.

**Proteomics**. GSC lines G561, G564, and G583 were cultured adherently for 6 days with 1 µM of GSK591, 1 µM LLY283, or the inactive control, SGC2096 (*n* = 3). Cells were subsequently lysed in denaturing urea buffer containing protease inhibitors and phosphatase inhibitors. Proteins were extracted and then digested using trypsin. After trypsin digestion, peptides were desalted and analyzed by one-dimensional separation MS using LC gradients on the Orbitrap Fusion Mass Spectrometer. Peptide identification, protein inference, and label-free quantification was performed using PEAKS (version X). The relative abundance of all peptides for each protein was considered for quantification. Subsequent data analysis was done in R (version 3.4.4). The MS proteomics data have been

deposited to the ProteomeXchange Consortium (http://proteomecentral.proteomexchange.org) via the PRIDE partner repository[64] with the dataset identifier PXD021635 and DOI: 10.6019/PXD021635.

**Proteomics and RNA-seq correlation**. For the 5355 proteins detected in one or more samples treated with PRMT5 inhibitor compounds LLY283 and GSK591 or the inactive compound SGC2096, the average $\log_2$ fold change in total peptide abundance between active and inactive compound-treated samples was calculated. Undetected proteins were assigned a $\log_2$ expression value of 0. UniProt accession numbers were converted to official gene symbols using the mapping provided by HGNC to identify overlaps with alternatively spliced genes as detected by RNA-seq. Pearson correlation between fold change expression from RNA-seq and proteome was performed in R.

For validation of splicing observed by RNA-seq, genes affected by an ASE considered significant in at least two out of the three cell lines used (684 genes, including 194 detected proteins) or the subset of these ASEs predicted to be disruptive (282 genes, including 70 detected proteins) were considered. Fold enrichment of proteins affected by an ASE within the 50 and 500 most downregulated proteins after PRMT5 inhibition was calculated. Fisher's exact test was used to determine statistical significance of the enrichment within downregulated proteins and standard error was estimated by bootstrapping with 100 re-samplings.

**AS-PCR and capillary electrophoresis**. Total RNA was extracted with the RNeasy Mini Kit (Qiagen, 74104). Contaminating genomic DNA was eliminated with on-column DNase digestion step using the RNase-free DNase set (Qiagen, 79254). RNA integrity and quality was assessed with an Agilent 2100 Bioanalyzer (Agilent Technologies)[65]. In all, 1.1 μg total RNA was reverse transcribed in a total volume of 20 μL using Transcriptor reverse transcriptase in the presence of random hexamers, dNTPs (Roche Diagnostics), and 10 units of RNAseOUT (Invitrogen) following the manufacturer's protocol. All primers were individually resuspended in Tris-EDTA buffer (IDT) to 100 μM stock solution and further diluted as a primer pair to 1.2 μM in RNase-free water (IDT). All the primers and expected amplicon sizes are listed in Supplementary Table 3. End-point PCR reactions were performed using 10 ng cDNA in the presence of 0.2 mM each dNTP, 1.5 mM $MgCl_2$, 0.6 μM each primer, and 0.2 units of Platinum Taq DNA polymerase (Thermo Scientific) in a final volume 10 μL. An initial denaturation step of 2 min at 95 °C was followed by 35 cycles at 94 °C 30 s, 55 °C 30 s, and 72 °C 60 s. The amplification was completed by a 2-min elongation step at 72 °C. PCR reactions were performed on SimpliAmp thermocyclers PCR System (Applied Biosystems, Life Technologies). The amplified products were resolved and analyzed via automated, chip-based microcapillary electrophoresis on Labchip GX Touch HT instruments (Perkin Elmer). Amplicon sizing and relative quantitation was performed using the manufacturer's software.

In alternative splicing RT-PCR assays, we are monitoring the expression shift between different isoforms of an ASE. PSI is a metric based on the long and short isoforms, respectively, representing the alternative inclusion and exclusion of an ASE. PSI is calculated as $L/(L + S)$, where $L$ and $S$ are the measured expression of the long and short isoforms, respectively. Thus PSI is interpreted as the expression portion taken by the event inclusion part. PSI is reported as a ratio value between 0 and 1, or as a percentage.

PSI is defined using two isoforms. When there are more than two amplicons detected, the PSI algorithm automatically associates long and short isoforms to the two most abundantly expressed amplicons in all assays for each ASE (primer pair).

**In vivo transplantation for orthotopic xenografts**. For intracranial xenograft experiments, $2 \times 10^4$ GSCs (G411) were injected into the forebrain of female NSG mice. The NSG mice were anesthetized using isoflurane (gas) and immobilized in a stereotaxic head frame. An incision was made at the midline of the skull and a hole was drilled 2 mm lateral and 1 mm posterior to the bregma using a 21-gauge needle. Cells were injected in 2 μL volume, 3 mm deep into the skull using a 26-gauge needle and Hamilton syringe. Syringe was held in place and the cells were slowly injected over 2 min. The needle was withdrawn gradually, over 3 min to avoid reflux. The hole was filled with bone wax and the incision sutured. Mice were monitored for health twice a day for 3 days after surgery.

**Pharmacokinetics (PK) of LLY-283**. LLY-283 was weighed and dissolved in enough DMSO to make 10 times (10×) the required concentration. 10× LLY-283 in DMSO was formulated in 1% HEC/0.25% Tween 80/0.05% antifoam before administration. This 10× LLY-283 solution can be incubated in a water bath for 5 min if the mixture is not clear. NSG mice were treated with 25, 50, or 100 mg/kg of LLY-283, administered through oral gavage, and sacrificed after 24 h. Blood was collected in heparinized tubes at end point by cardiac puncture and plasma was separated by centrifugation. LC-MS was used to determine the plasma concentrations of LLY-283. To determine brain penetration, transcardiac perfusion with phosphate-buffered saline was performed, after which whole brains were harvested for the determination of concentration of LLY-283. Perfused brains were

flash frozen in liquid nitrogen and stored at −80 °C. LLY-283 concentration in the brain was analyzed using LC-MS. Concentration of LLY-283 in the blood and brain was quantified using a linear regression standard curve derived from 10 concentrations of LLY-283 ranging from 10 to 10,000 ng/mL.

**Tolerability study of LLY-283**. NSG mice were administered 50 mg/kg of LLY-283 (3 mice per group) using two different regimens, namely, weekly cycles of 3 days of treatment followed by a 4-day break (Regimen 1) or weekly cycles of 5 days of treatment followed by a 2-day break (Regimen 2). Weight and condition of mice were recorded twice a week. Mice who had lost >20% of their body weight were sacrificed.

**In vivo LLY-283 treatment**. Twenty thousand G411 cells were orthotopically transplanted into NSG mice using the method described above (12 mice per treatment group, total of 24 mice). LLY-283 (50 mg/kg) or vehicle (1% HEC/0.25% Tween 80/0.05% antifoam) was administered to the mice through oral gavage 1 week after tumor cell implantation. LLY-283 or vehicle was administered daily for 3 days, followed by a 4-day treatment interval, for a total of 4 cycles or until end point (death). Mice were euthanized when neurological symptoms were observed along with a domed head. Two mice from the vehicle group were censored from the study: one due to ectopic growth of the tumor (in the neck) and one due to late tumor development. Survival analysis was performed using the GraphPad Prism 7 software. Kaplan–Meier curves show time elapsed (in days) from tumor transplantation to death ($p > 0.05$). Statistical analysis was performed using log-rank (Mantel–Cox) test.

**Immunohistochemistry**. Upon sacrifice, mouse brains were removed and immediately fixed in 10% formalin and then washed in 70% ethanol before embedding in paraffin. Five-μm coronal sections were cut for H&E staining. H&E staining was carried out according to the manufacturer's instructions (MHS32-1L, Sigma-Aldrich and 6766009, Thermo Scientific). Images were acquired using a 3DHistech Pannoramic 250 Flash II Slide Scanner and analyzed using the Pannoramic Viewer software (3DHISTECH).

**RNA sequencing**. GSC lines G561, G564, and G583 were cultured adherently for 3 days with either 1 μM of GSK591 or the inactive control, SGC2096. RNA was extracted using the RNeasy Mini Kit (Qiagen, 74104) with on-column DNase digestion using the RNase-free DNase set (Qiagen, 79254). Two micrograms of RNA was sequenced using the HiSeq 2500 (Illumina) on a high-throughput flow cell with V4 chemistry. Sequencing was done as paired end reads with a read length of 126 bases. Raw sequencing information was collected in the form of FASTQ files. RNA-seq libraries were constructed using the NEBNext Ultra Directional RNA Library Prep Kit (NEB, #E7760), following the manufacturer's instructions. The raw trimmed reads were aligned to the reference genome UCSC hg19. DESEQ2 was used for differential expression, with the Sample ID and treatment taken as covariates in the model. Differential expression was assessed as LFC between PRMT5 inhibitor treated (numerator) and untreated (denominator) samples, after taking into account sample effects. Adjusted $p$ value was calculated by BH correction. A ranked gene list for GSEA was constructed with genes ranked by the following statistic: sign (LFC)*(−log10(adjusted p.value)). Any genes that obtained infinity scores were set to sign(LFC)*max(abs(non.inf)) + sign(LFC)*runif(0,1), where max(abs(non.inf)) represents the maximum absolute value of non-infinite ranking scores, and runif(0,1) represents a number chosen at random from a uniform(0,1) distribution. Pathway and gene ontology (GO) analyses were performed using GSEA. Pathways included all GO annotations (GO:BP, GO:MF, GO:CC), excluding I.E.A. (inferred by electronic annotation), as well as several other pathway databases such as REACTOME and Msigdb. The file used was Human_GO_AllPathways_no_GO_iea_October_01_2018_symbol.gmt. Further details are available from http://baderlab.org/GeneSets. Pathways were filtered by size (15 ≤ gene set size ≤ 200), and GSEA was run with 1000 gene set permutations. GSEA results were visualized in Cytoscape v.3.6.1 using the Enrichment Map application. Pathways were filtered at FDR 0.1 and default settings for GSEA analysis type. Autoannotate was run with default settings. For classification of GSC lines according to TCGA subtypes, Log2 transformed fpkm values were batch corrected using the Combat function in the SVA package[66]. Log2 transformed fpkm values were fed into GSVA[67], scoring for the four TCGA subtypes found by Verhaak et al.[27]. Individual cell lines were classified as a Proneural, Classical, Neural, Mesenchymal, or combination thereof, based on GSVA scores that were found to be significantly associated with that sample. Subtype classifications were ordered from top to bottom in order of decreasing GSVA enrichment score.

**Splicing data analysis**
*Alignment reading, transcript reconstruction, and ASE identification*. The sequencing quality for each library was evaluated using FastQC[68] v0.11.8. Subsequent alignment to human genome GRCh38.p5 was performed with STAR aligner v2.6.1[69], using v24 from GENCODE's[70] comprehensive gene annotation as a reference. Resulting BAM files were evaluated with samtools flagstat[71] to ensure

that most of the reads aligned were uniquely mapped and properly paired. Unless specifically stated, all quality assessment, filtering, treatment, alignment, and post-processing of the data was performed in accordance with ENCODE's guidelines and best practices for RNA-seq of human samples. Aligned files from all samples were used as input for the Stringtie[72] v.1.3.4 pipeline for the assembly of transcripts found in the RNA-seq data. The resulting merged gene transfer format file was compared with GENCODE v24 comprehensive annotation for the identification of known and novel transcripts. The pipeline was used with the recommended parameters for human samples. ASEs for four classes (exon skipping, intron retention, MXEs, and alternative 3' and 5' splice sites) were identified using MATS[73] v3.2.5 in a single-sample mode. Intron retention events were identified using IRFinder[74] v.1.2.4 for single samples. For all classes, statistically significant events were selected based on a combination of parameters: $p$ value ≤0.05; at least 1 read supporting the inclusion event; at least 1 read supporting the exclusion event; at least 10 total reads supporting the event; and ΔPSI (Treated − Control) ≥0.10.

*ASE characterization.* Genes containing ASEs were also analyzed for enrichment of GO classes (cellular component, biological process and molecular function), PFAM domains, InterPro domains, KEGG Pathways, Tissue and Disease associations; these analyses were performed using STRINGdb[75] v.10.5. Additionally, alternatively spliced events were cross-examined with VastDB[35] (hg38 assembly) using bedtools intersect to identify common elements based on their genomic coordinates, requiring a 95% reciprocal overlap and same strandness to be considered the same event. Events with corresponding matches in VastDB were analyzed for predicted protein impact, while events without a corresponding match in VastDB and/or with "Unknown" protein impact were mapped to reconstructed transcripts from RNA-seq data (from StringTie) with the following criteria: for RIs, we selected transcripts in which the RI was completely inserted between two exons in a full-length transcript (complete inclusion); for skipped exons (SE), we selected transcripts in which the SE was completely excluded from the full-length transcript and completely inserted in at least one reference (canonical) transcript; for included exons (IE), we selected transcripts in which the IE was completely inserted into the full-length transcript and completely excluded from at least one reference (canonical) transcript. Identified transcripts were cross-referenced with GENCODE annotated biotypes for identification of these alternative transcripts, with unannotated transcripts labeled as "stringtie_transcript". These transcripts (excluding those with "protein_coding" biotype) were subsequently characterized for their coding potential and mRNA/ORF size ratio using CPAT[76] v.1.2.4.

*Generating splicing signature from baseline RNA-seq data.* ASEs were selected based on three criteria. First, the events must accurately be quantifiable in at least 75% of the samples; the intra-group variation must be smaller than the inter-group variation (as quantified by standard deviation); and finally, the event must have a mean PSI difference of at least 20% between the groups (good responders vs poor responders). Selected events were also mapped against VastDB (using their respective genomic coordinates; bedtools intersect -f 0.95 -r True -s True) to identify annotated and discriminate potentially novel events.

**Reporting summary**. Further information on research design is available in the Nature Research Reporting Summary linked to this article.

## Data availability

RNA-seq and WGS data for GSC lines reported in this manuscript have been deposited at the European Genome-phenome Archive, EGA study ID EGAS00001004395. The RNA-seq raw data files for the three GSC lines treated with GSK591 or SGC2096 are available on EGA under the accession ID EGAS00001004397. Since these are patient-derived samples, access is controlled. Parties wishing to access the data should fill out the "Data Access Agreement form" provided by EGA. All legitimate requestors affiliated with an academic institution or industry will be granted access from the corresponding author as long as they sign a form saying that they will not use the data to attempt to identify the patient it came from. This is standard practice in order to meet REB requirements to uphold patient privacy laws. The RT-PCR data with capillary electropherograms and PSI calculations can be accessed here: https://rnomics-store.med.usherbrooke.ca/palace//data/related/3289. The mass spectrometry proteomics data have been deposited to the ProteomeXchange Consortium via the PRIDE partner repository with the dataset identifier PXD021635. The raw GSK591 and LLY-283 dose–response data are available through https://github.com/bhklab/PRMT5i_GBM. Source data are provided with this paper.

## Code availability

The code used to calculate AACs using the Incucyte confluence data is available through https://github.com/bhklab/PRMT5i_GBM.

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

## Acknowledgements

We thank Richard Moore, Yussanne Ma, Andrew Mungall, Marco Marra and the Michael Smith Genome Sciences Centre for the WGS and RNA-seq of the GSC samples. We would like to thank the SickKids-UHN Flow Cytometry Facility, the RNomics platform at Université de Sherbrooke, the OICR BioLab, the Lunenfeld-Tanenbaum Research Center/Mt Sinai high-content screening facility and the Princess Margaret Bioinformatics group for contributions to this work. We thank Suzanne Ackloo and Peter Brown for providing the chemical probes and controls, Lihua Liu and Matthieu Schapira for data analysis and visualization tools, Misha Bashkurov and Brigitte Theriault for help with HCS image analysis, Qin Wu for preparing the vehicle, Graham McLeod and Stephane Angers for sharing CRISPR screen data, Kristen Jensen-Pergakes for the LLY-283 compound and for insights into the work, and Tracey Richards for administrative support. We thank all members of the Dirks laboratory and Structural Genomics Consortium (SGC), Toronto for helpful and stimulating scientific discussions. Research was supported by SU2C Canada Cancer Stem Cell Dream Team Research Funding (SU2C-AACR-DT-19-15) provided by the Government of Canada through Genome Canada and the Canadian Institute of Health Research (142434), with supplemental support from the Ontario Institute for Cancer Research through funding provided by the Government of Ontario. Stand Up To Cancer Canada is a program of the Entertainment Industry Foundation Canada. Research funding is administered by the American

Association for Cancer Research International-Canada, the scientific partner of SU2C Canada. We thank the Hospital for Sick Children Foundation, Jessica's Footprint, Hopeful Minds Foundation, and B.R.A.I.N. Child. P.B.D. holds a Garron Family Chair in Childhood Cancer Research at The Hospital for Sick Children. P.S. is supported by a MITACS Accelerate fellowship. K.B.M received financial support through FAPESP (grants 2012/0195-3, 2014/50897-0, and 2014/21704-9). F.E.C. was a recipient of a FAPESP fellowship (2016/25521-1, 2015/25134-5) and supported by the SGC Scientist exchange program. The SGC is a registered charity (number 1097737) that receives funds from AbbVie, Bayer Pharma AG, Boehringer Ingelheim, Canada Foundation for Innovation, Eshelman Institute for Innovation, Genome Canada through Ontario Genomics Institute [OGI-055], Innovative Medicines Initiative (EU/EFPIA) [ULTRA-DD grant no. 115766], Janssen, Merck KGaA, Darmstadt, Germany, MSD, Novartis Pharma AG, Ontario Ministry of Research, Innovation and Science (MRIS), Pfizer, São Paulo Research Foundation-FAPESP, Takeda, and Wellcome.

## Author contributions

Conceptualization, P.S., P.P., P.B.D., C.H.A.; methodology, P.S., P.P., P.B.D., C.H.A., Z.Q.B.; formal analysis, W.B.-A., F.M.G.C., P.G., F.E.C., L.M.R., O.W., M.S.-O., J.C.-H., P.T., B.-C.D.; investigation, P.S., J.H., V.V., M.S.O., E.K, A.M.A., P.G., L.L., M.M.K., H.W., N.R., X.L., X.C., M.M., N.T., J.V., K.H., I.R., O.Z., M.D., W.C., F.J.C; resources, P.B.D., C.H.A., D.B.-L., Z.Q.B., S.W., M.B., M.D.D., S.D., J.S.; writing, P.S., P.B.D., C.H.A., P.P., J.C.H.; project administration, F.J.C.; supervision, P.B.D., C.H.A., P.P., B.H.-K., K.B.M., G.D.B., T.J.P., M.L., H.A.L., S.W., A.C., H.R., M.T.; funding acquisition, P.B.D., C.H.A.

## Competing interests

Z.Q.B. is an employee of Eli Lilly and Company. All other authors declare no competing interests.

## Additional information

[1]Developmental and Stem Cell Biology Program and Arthur and Sonia Labatt Brain Tumor Research Centre, The Hospital for Sick Children, Toronto, ON, Canada. [2]Structural Genomics Consortium, University of Toronto, Toronto, ON, Canada. [3]Center for Molecular Biology and Genetic Engineering, University of Campinas (UNICAMP), Campinas, Brazil. [4]The Structural Genomics Consortium, University of Campinas (UNICAMP), Campinas, Brazil. [5]Princess Margaret Cancer Centre, University Health Network, Toronto, ON, Canada. [6]Department of Medical Biophysics, University of Toronto, Toronto, ON, Canada. [7]Institute for Research in Immunology and Cancer, Université de Montréal, Montreal, QC, Canada. [8]RNomics Platform, Université de Sherbrooke, Sherbrooke, QC, Canada. [9]Ontario Institute for Cancer Research, Toronto, ON, Canada. [10]The Donnelly Centre, University of Toronto, Toronto, ON, Canada. [11]Department of Molecular Genetics, University of Toronto, Toronto, ON, Canada. [12]Department of Pharmacology and Toxicology, University of Toronto, Toronto, ON, Canada. [13]Hotchkiss Brain Institute, Cumming School of Medicine, University of Calgary, Calgary, AB, Canada. [14]Department of Cell Biology and Anatomy, University of Calgary, Calgary, AB, Canada. [15]Maple Flavored Solutions, LLC, Stony Brook, NY, USA. [16]Lilly Research Laboratories, Eli Lilly and Company, Indianapolis, IN, USA. [17]Division of Neurosurgery, Department of Surgery, University of Toronto, Toronto, ON, Canada. [18]Division of Neurosurgery, Toronto Western Hospital, University Health Network, Toronto, ON, Canada. [19]Division of Neurosurgery, Department of Surgery, St. Michael's Hospital, Toronto, ON, Canada. [20]The Arthur and Sonia Labatt Brain Tumour Research Centre, The Hospital for Sick Children, Toronto, ON, Canada. [21]Department of Medical Imaging, St. Michael's Hospital, University of Toronto, Toronto, ON, Canada. [22]Department of Computer Science, University of Toronto, Toronto, ON, Canada. [23]Vector Institute, Toronto, ON, Canada. [24]Clark H. Smith Brain Tumor Centre, Cumming School of Medicine, University of Calgary, Calgary, AB, Canada. [25]Department of Laboratory Medicine and Pathobiology, University of Toronto, Toronto, ON, Canada. [26]Division of Neurosurgery, The Hospital for Sick Children, Toronto, ON, Canada. ✉email: takis.prinos@utoronto.ca; cheryl.arrowsmith@uhnresearch.ca; peter.dirks@sickkids.ca

