## [Peer Review File · Nature Communications]

Reviewers' comments:

Reviewer #1 (Remarks to the Author); expert in GSC and mouse models:

In this manuscript, the authors, Sachamitr, P. et al., performed a screening for growth inhibition on a total of 46 patient-derived glioma stem-like cell (GSC) lines by using the epigenetic probes, inhibitors of major histone modifying enzymes and related proteins. Among three effective inhibitors, two are inhibitors for PRMT5. The authors selected three or four responsive GSC lines and further characterized the responses and molecular coordinates in these GSCs. The effects of inhibition of PRMT5 on symmetrical dimethylation of arginine in proteins, gene expression and RNA splicing was assessed and cell cycle-regulation genes were among the major targets. Finally, one of the PRMT5 inhibitor, LLY-283 was found to inhibit GSC cell growth and intracranial GSC tumor xenograft in animals. This is an interesting and straightforward study with potential significance. The methods and approaches as well as the techniques employed in this study are excellent. The data presentation is of high quality and supportive to the conclusion. However, the overall research design is limited and the major information derived from this study only provides incremental advance for our current understanding. Data of functional validation in vivo study is also disappointing. The current enthusiasm for this study to be considered for publication in Nature Communications is minimum.

Major Comments:

1. The rationale of selection of GSC lines as model system for characterizing PRMT5 inhibitors is lacking. From the Table S2, IC50 of two inhibitors shows marked differences among the 46 GSC lines. Three responsive GSC lines have excellent low IC50 values for both inhibitors. However, some of GSC lines had extremely high IC50 values for these inhibitors. For example, for GSK591, G549 has IC50 of 144, BT89 has IC50 of 1499 and BT284 has 591. The underlying mechanism for these resistant GSC lines was not investigated in this study.
2. PRMT5 function has been linked to status of MTAP deletion as well as co-deletion with CDKN2A/2B in various types of human cancers including GBM. In Figure 2A and 2C, the authors tried to exclude this link by showing that various GSC lines displayed different responses to the inhibitory effects using AAC values. However, the key data in Figure 2C does not include any of the four GSC lines that were used as the sole model system in this study. GSC411, GSC561, GSC583, and GSC837 are not included in the figure 2C! Additionally, the authors must provide convincing genetic and biochemical data to demonstrate the deletion and expression of MTAP, CDKN2A/2B genes and proteins in all GSC lines.
3. Genetic approaches must be used to support the small molecule inhibition on GSC biologic properties. For example, if the cell cycle genes are the major effectors for PRMT5 inhibitors' effects, the genetic modulation of the major targeting genes should either enhance or resist to the inhibition by the PRMT5 inhibitors. Same applies to effects of mRNA splicing.
4. In Figure 3f, what are those splicing events that correlate to which genes?
5. In vivo data using only one GSC model is insufficient. The in vitro effects by the inhibitors must be validated in vivo GSC tumor xenografts. These can also be assessed using tumor xenografts in animals that eventually succumbed to the tumor.
6. The rationale of responsiveness to these two inhibitors by human fetal neural stem cells should be investigated. Did authors assess responsiveness to these inhibitors by adult human neural stem cells, neurons, and astrocytes?

Reviewer #2 (Remarks to the Author); expert in alternative splicing:

This is an interesting manuscript that identifies PRMT5 as a promising therapeutic target for glioblastoma using patient derived cell lines and two distinct PRMT5 inhibitors. While prior publications have identified PRMT5 as an important therapeutic target in this disease and the impact of PRMT5 inhibition on splicing in this disease, these data are an important extension of prior work. There are two points that should be dealt with to improve the quality of information presented as follows:

-Why are glioblastoma cells preferentially sensitive to PRMT5 inhibition than normal astrocytes? Is this related to differences in cell proliferation or uptake of drug?

-The authors suggest that "different splicing biology" may underlie differences in responses of glioma cells to PRMT5 inhibition. This phrase is inappropriately vague and a deeper biological evaluation of this difference should be provided. How do the authors know that there is a difference in splicing biology as opposed to differences in arginine methylation/demethylation or metabolism of proteins which undergo arginine methylation/demethylation? Are there differences in RNA splicing in these cell lines prior to drug exposure? Are there differences in SDMA levels upon PRMT5 exposure across these different cell types? Differences in drug uptake? Some evaluation of splicing changes induced by PRMT5 inhibition beyond representation of data from RNA-seq should be shown as well (e.g. isoform specific RT-PCR in cells with differential response to drug).

Response to Reviewers of Sachamitr *et. al.* “Disruption of splicing, proliferation and stemness via PRMT5 inhibition as a therapeutic strategy for Glioblastoma”

We thank the reviewers for their constructive comments. We have performed extensive new experiments contributing to major revisions to the paper to address the reviewers' comments. Please see below for a detailed list of our responses (in italics) to the reviewer's comments with reference to accompanying changes in the manuscript.

Reviewer #1, Major Comments:

-Reviewer's comment #1: The rationale of selection of GSC lines as model system for characterizing PRMT5 inhibitors is lacking. From the Table S2, IC50 of two inhibitors shows marked differences among the 46 GSC lines. Three responsive GSC lines have excellent low IC50 values for both inhibitors. However, some of GSC lines had extremely high IC50 values for these inhibitors. For example, for GSK591, G549 has IC50 of 144, BT89 has IC50 of 1499 and BT284 has 591. The underlying mechanism for these resistant GSC lines was not investigated in this study.

Response: *We appreciate that the rationale for choosing specific GSC lines for deeper investigation of drug response and mechanism was not well articulated, nor as systematic as it might have been in our original manuscript. In this revised manuscript we have highlighted five GSC cell lines that span both poor responder (G729, G797) and good responder lines (G411, G561, G583), now highlighted with asterisk in Figure 2A. We selected both poor and good responders specifically to try to understand the factors that mediate response to PRMT5 inhibition.*

*In order to rule out resistance mechanisms related to exposure to the chemical probes, we have now performed Western blot analysis for the SDMA methyl mark in exemplar responsive and non-responsive GSC lines after treatment with GSK591 or LLY-283 or inactive control (data added in **Figure 1E, and supplementary Figure S1E**). The results clearly show that GSK591 and LLY-283 both decreased the SDMA mark equally well in both 'responder' and 'non-responders' GSC lines. These data show that the target enzyme is being effectively inhibited in all cases. Therefore, the observed differences in response are likely due to molecular features unrelated to typical resistance mechanisms such as cell-specific action on the chemical probes, such as efflux, metabolism or compound stability.*

As the reviewer points out, and as often seen for drug treatment of other cancer cell lines and patients (e.g. PMID: [29946150](https://pubmed.ncbi.nlm.nih.gov/29946150/)), we indeed observed a wide range of responses to PRMT5 inhibition. This familiar 'waterfall plot' is not unexpected and provides a picture of the variability of responses that may be expected in the clinical setting across multiple patients. While we did not focus our characterization on resistance mechanisms per se, our study sought to identify trends across these GSC cell lines that could distinguish good vs poor responders to PRMT5i. Importantly, we have shown that in these low-passage patient-derived GSC lines, all of the previously purported markers of response (MTAP or p53 status, MTA levels,) identified in other tissue types, or in long-passaged glioma cell lines (the ratio of CLNS1A to RIOK1), do not hold true across our well-annotated set of GSC lines. Instead, we identified a novel pre-existing splicing signature that can distinguish the best from the worst responders to PMRT5 inhibition.

-Reviewer's comment #2: PRMT5 function has been linked to status of MTAP deletion as well as co-deletion with CDKN2A/2B in various types of human cancers including GBM. In Figure 2A and 2C, the authors tried to exclude this link by showing that various GSC lines displayed different responses to the inhibitory effects using AAC values. However, the key data in Figure 2C does not include any of the four GSC lines that were used as the sole model system in this study. GSC411, GSC561, GSC583, and GSC837 are not included in the figure 2C! Additionally, the authors must provide convincing genetic and biochemical data to demonstrate the deletion and expression of MTAP, CDKN2A/2B genes and proteins in all GSC lines.

Response: *While original Figure 2C did include lines G583 and G837, we agree that it was not sufficiently comprehensive. We have therefore performed extensive new experiments providing genomic data for all the GSC lines in the new **panel 2B**. This includes copy number status for MTAP and CDKN2A/2B genes. In addition, we added a new **panel C** to **Figure 2** showing MTAP and CDKN2A protein levels by Western blot across the GSC line cohorts. Furthermore, we have added a new panel in **Supplementary Figure 2C** showing a heatmap of methylthioadenosine (MTA) levels across a panel of GSC lines as measured by mass spectrometry. We modified the text accordingly to reflect these changes (see page 12).*

-Reviewer's comment #3: Genetic approaches must be used to support the small molecule inhibition on GSC biologic properties. For example, if the cell cycle genes are the major effectors for PRMT5 inhibitors' effects, the genetic modulation of the major targeting genes should either enhance or resist to the inhibition by the PRMT5 inhibitors. Same applies to effects of mRNA splicing.

Response: *Our data indicate that the cell cycle effect is combinatorial with changes to expression of a cohort of over 14 cell cycle protein isoforms (due to splicing changes). In our revised manuscript we provide numerous literature citations that link individual altered protein isoforms observed in our study with previously noted effects on cell cycle, tumor suppression or growth regulation. We note that the altered splicing events (ASEs) do not change the target proteins to a single isoform, but rather the ASEs affect the proportion of protein isoforms produced for a large number of target proteins. Thus, knocking out any single gene or isoform is unlikely to reproduce or rescue the PRMT5 inhibition phenotype.*

-Reviewer's comment #4: In Figure 3F, what are those splicing events that correlate to which genes?

Response: *Our original Figure 3, panel F included a heatmap with 45 splice variants that we found to be differentially spliced between good and poor responders across a panel of 33 GSCs. To make this point more clear, we added the gene names to this heatmap and made a new separate **Figure 4** showing the predictive biomarker ASEs heatmap with gene names indicated (**Figure 4**). We also included the genomic coordinates and PSI details for each ASE in **Table ST3**. Importantly, we see an enrichment for genes involved in cytoskeletal processes like ciliogenesis among these splice variants suggesting structural differences between PRMT5i responding and non-responding GSC lines. These points are now discussed on page 32.*

-Reviewer's Comment #5: In vivo data using only one GSC model is insufficient. The in vitro effects by the inhibitors must be validated in vivo GSC tumor xenografts. These can also be assessed using tumor xenografts in animals that eventually succumbed to the tumor.

Response: We established new orthotopic xenograft models for two additional GSC lines and treated them with the chemical probe LLY-283, but the results are inconclusive due to technical issues. The G411 GSC line, reported in our original manuscript, is one of our fastest growing lines, and has been used for other studies in the Dirks lab [PMIDs: 27300435, 26626085]. However, the two new GSC lines (G837 and G523) are slower growing compared to G411, and required longer treatment times with LLY-283. In the G523 xenograft model the treated mice became sick (exhibited weight loss and scruffy appearance) and had to be sacrificed before end-point, thus precluding us from assessing a survival advantage. For the G837 model we were able to complete the experiment, however, only a very slight trend toward survival advantage was observed. Notably, the Western blots of the SDMA mark in treated vs untreated tumour of this model showed no appreciable SDMA decrease in the LLY-283 treated group, making these results also inconclusive. The annotated survival curves are shown below.

We note that LLY-283 is a tool compound intended primarily for cellular studies, and not a drug candidate with optimized PK/PD properties. As noted in our brief PK studies (Page 26) continuous dosing of LLY-283 is not tolerated by NSG mice requiring an interleaved 3-days on/4-days off dosing schedule, and even so, some modest weight loss was seen (10%). Given these observations, we suspect that the apparent toxicity observed in the treated mice of the G523 model may be due to the longer treatment with LLY-283. We feel that further attempts at optimizing in vivo conditions for chemical probe are not the best use of mice or resources,

especially given the long timelines for these slow growing tumors and given our lab shutdown during the Covid19 pandemic.

We feel that the robust nature of our data characterizing the splicing effects of PRMT5 and mapping of the molecular and phenotypic effects across a larger cohort of patient samples will inspire pharmaceutical or drug discovery teams to develop brain penetrant PRMT5 inhibitors for the study and treatment of gliomas.

-Reviewer's comment # 6: The rationale of responsiveness to these two inhibitors by human fetal neural stem cells should be investigated. Did authors assess responsiveness to these inhibitors by adult human neural stem cells, neurons, and astrocytes?

Response: *Yes, the response of human neural stem cells to these two inhibitors was already shown in **Suppl. Figure 1C** and astrocytes in **Suppl. Figure 1B** of the previously submitted manuscript. In addition, we have now added a panel in **Suppl. Figure 1E** with Westerns of the SDMA mark in human fetal neural stem cells (HFNS) and normal human astrocytes (NHAs) after treatment with GSK591 or LLY283. These Western results show clearly that the SDMA mark is decreased quite dramatically by either PRMT5 inhibitor in all these lines. We believe these results adequately address both the reviewer's comments about differences in SDMA levels across different cell types and responsiveness to the two PRMT5 inhibitors in normal neural stem cells and astrocytes.*

Reviewer #2.

-Reviewer's comment: Why are glioblastoma cells preferentially sensitive to PRMT5 inhibition than normal astrocytes? Is this related to differences in cell proliferation or uptake of drug?

Response: *This was addressed for Reviewer 1's comments #1 & #6 (see above). Briefly, we rule out differences in drug uptake, efflux or stability between cell lines as the levels of the SDMA mark were equally decreased in treated astrocytes or GSCs (**see Suppl. Figure 1E**). We cannot rule out differences in cell proliferation rates as human astrocytes seem to proliferate much slower (**Suppl. Figure 1B**) than GSCs lines (**Suppl. Figure 1A**).*

-Reviewer's comment: The authors suggest that "different splicing biology" may underlie differences in responses of glioma cells to PRMT5 inhibition. This phrase is inappropriately vague and a deeper biological evaluation of this difference should be provided. How do the authors know that there is a difference in splicing biology as opposed to differences in arginine methylation/demethylation or metabolism of proteins which undergo arginine methylation/demethylation?

Response: *The reviewer raises some very good points. Regarding differences in arginine methylation following treatment with PRMT5 inhibitors, we assessed the levels of symmetric dimethyl Arginine (SDMA) across GSC lines with different responses (**Figure 1E and Supplemental Figure S2G**). Our results show no differences in SDMA levels after treatment with GSK591 or LLY-283 across 6 GSC lines. However, we cannot exclude the existence of other differences in Arginine de/methylation or metabolism of proteins. We therefore have removed the phrase 'different splicing biology' from the text. What we intended with this phrase,*

and as elaborated below and in the revised manuscript, is that there is evidence of inherent differences in the pre-existing splicing isoforms between good and poor responders.

-Reviewer's comment: Are there differences in RNA splicing in these cell lines prior to drug exposure?

Response: Yes, we added a separate Figure 5 with a heatmap summarizing the splicing differences (represented as PSI values) derived from bioinformatics analysis using RNA-Seq data across a panel of 33 untreated GSC lines (**Figure 5A**). These splicing changes can discriminate between good responders and poor responders to PRMT5 inhibitors (Figure 5A). We added a detailed section in the discussion with specific examples for the most prominent ASEs with known biology that help explain our observed differential phenotypes.

-Reviewer's comment: Are there differences in SDMA levels upon PRMT5 exposure across these different cell types? Differences in drug uptake?

Response: No, see response to reviewer's #1, point #1 above. Briefly, Westerns of the SDMA mark did not show differences in SDMA levels across responders and non-responders. These data also allow us to rule out differences in drug uptake, metabolism, etc. between GSC lines.

-Reviewer's comment: Some evaluation of splicing changes induced by PRMT5 inhibition beyond representation of data from RNA-seq should be shown as well (e.g. isoform specific RT-PCR in cells with differential response to drug).

Response: Yes, we did RT-PCRs for a selection of 14 ASEs based on the RNA-Seq analysis which were identified as differentially spliced after treatment with either GSK591 or LLY-283. A summary of the splicing changes (represented as PSI values) is shown as a heatmap in **Figure 3F**. We added the gel images of the RT-PCRs for one GSC line (G583) in supplemental **Figure S4**. Detailed PSI values and amplicon sizes for each gene isoform across 6 GSC lines are provided in **Table ST4**. In addition, we have created an interactive online version that allows for better visualization and interrogation of the RT-PCR data showing individual reactions for each individual ASE (gene isoform) across the different cell lines and treatments with bar graphs, heatmaps and capillary electropherograms at <https://rnomics-store.med.usherbrooke.ca/palace//data/related/3289>. Access is currently password protected but will become publicly available upon acceptance of the manuscript. Username: sachamitr2019@suppl.data; password: Sachamitr2019SD). We also added a section in the text (pages 16-17) to reflect these changes. Furthermore, we added global proteomics data derived from the same 3 GSC lines we had done RNA-Seq treated with GSK591 or LLY-283 which showed an enrichment for downregulated proteins among the disruptive ASEs (Figure 3E). This provides further functional support for the disruptive splicing effect on the proteome. We have added additional sections describing these results (pages 15-21) and methods describing the methodology used for this analysis.

REVIEWERS' COMMENTS

Reviewer #1 (Remarks to the Author):

The revised manuscript has addressed comments raised by this reviewer with new data and necessary revision in the text. The responses are satisfactory. The new data and data presentation are also excellent. Moreover, it is imperative that the authors do not include the excel file of table 3 and other tables in the pdf file of the supplemental materials. This made it impossible to read and assess the data. The current format of inclusion of tables in the pdf file make the file in very big size and not readable. The authors must re-submit the tables in separate excel file in the supplementary material.

Reviewer #2 (Remarks to the Author):

The authors have replied to my initial comments and concerns. I have no further issues with the manuscript.